

# Refreeze experiments of water droplets containing different types of ice nuclei interpreted by classical nucleation theory

Lukas Kaufmann[1], Claudia Marcolli[1,2], Beiping Luo[1], and Thomas Peter[1]

[1]Institute for Atmospheric and Climate Science, ETH, Zurich, Switzerland
[2]Marcolli Chemistry and Physics Consulting GmbH, Zurich, Switzerland

*Correspondence to*: C. Marcolli (claudia.marcolli@env.ethz.ch)

**Abstract.** Homogeneous nucleation of ice in supercooled water droplets is a stochastic process. In its classical description, the growth of the ice phase requires the emergence of a critical embryo from random fluctuations of water molecules between the water bulk and ice-like clusters, which is associated with overcoming an energy barrier. For heterogeneous ice nucleation on ice-nucleating surfaces both, stochastic and deterministic descriptions are in use. Deterministic (singular) descriptions are often favored because the temperature dependence of ice nucleation on a substrate usually dominates the stochastic time dependence, and the ease of representation facilitates the incorporation in climate models. Conversely, classical nucleation theory (CNT) describes heterogeneous ice nucleation as a stochastic process with a reduced energy barrier for the formation of a critical embryo in the presence of an ice-nucleating surface. This reduction is conveniently parameterized in terms of a contact angle $\alpha$ between the ice phase immersed in liquid water and the heterogeneous surface area. This study investigates various ice-nucleating agents in immersion mode by subjecting them to repeated freezing cycles to elucidate and discriminate the time and temperature dependences of heterogeneous ice nucleation. Freezing rates determined from such refreeze experiments are presented for Hoggar Mountain dust, birch pollen washing water and Arizona Test Dust (ATD) and nonadecanol coatings. For the analysis of the experimental data with CNT we assumed the same active site to be always responsible for freezing. Three different CNT-based parameterizations were used to describe rate coefficients for heterogeneous ice nucleation as a function of temperature, all leading to very similar results: for Hoggar Mountain dust, ATD and larger nonadecanol coated water droplets, the experimentally determined increase of freezing rate with decreasing temperature is too shallow to be described properly by CNT using the contact angle $\alpha$ as the only fit parameter. Birch pollen washing water and small nonadecanol coated water droplets show the reverse behavior with temperature dependencies of freezing rates steeper than predicted by CNT formulations. Good agreement of observations and calculations can be obtained when a prefactor $\beta$ is introduced to the rate coefficient as second fit parameter. Thus, the following microphysical picture





emerges: Heterogeneous freezing occurs on ice-nucleating sites that need a minimum (critical) surface area to host embryos of critical size to grow into a crystal. Fits based on CNT suggest that the critical active site area is in the range of $10 - 50$ nm$^2$ depending on sample, temperature, and CNT-based parameterization. Two fitting parameters are needed to characterize individual active sites. The contact angle $\alpha$ lowers the energy barrier that

has to be overcome to form the critical embryo on the site compared to the homogeneous case where the critical embryo develops in the volume of water. The prefactor $\beta$ is needed to adjust the calculated slope of freezing rate increase with decreasing temperature to the measured one. When it is large, there are many nucleation attempts and nucleation occurs immediately when the temperature is low enough so that the active site can accommodate a critical embryo. This is the case for active sites of birch pollen washing water and the small droplets coated with

nonadecanol. If the prefactor is low, the number of nucleation attempts is low and the increase of freezing rate with decreasing temperature is shallow. This is the case for Hoggar Mountain dust, the large droplets coated with nonadecanol, and ATD. Different hypotheses why the value of the prefactor depends on the nature of the active sites are discussed.

## 1 Introduction

Freezing of liquid droplets and subsequent ice crystal growth affects optical cloud properties and precipitation (IPCC, 2013). Field measurements show that ice nucleation in relatively warm cumulus and stratiform clouds may begin at temperatures much higher than those associated with homogeneous ice nucleation in pure water droplets. The glaciation of these clouds is ascribed to heterogeneous ice nucleation occurring on the foreign surfaces of ice-nucleating particles present in the cloud droplets. Ice nucleation induced by particles located within

the body of water or aqueous droplets is termed immersion freezing and is probably the most important nucleation process turning liquid droplets in relatively warm clouds into ice crystals (Murray et al., 2012).

Ice-nucleating surfaces are supposed to exhibit features or structures which promote ice nucleation. However, it is not clear whether these structures are extended over the whole surface or localized at specific sites. The concept of epitaxy considers an extended surface with a close lattice match to ice as responsible for ice nucleation.

Ice nucleation is assumed to occur at a random location on this uniform surface with a nucleation rate that scales linearly with surface area. However, there is increasing evidence that preferred locations present on surfaces are responsible for ice nucleation (e.g. Vali, 2014; Vali et al., 2015). Such sites are thought to be special surface regions such as crystal defects (Vonnegut, 1947), pores, cracks or ledges (Knight, 1979; Sear, 2011, Fletcher, 1969). If an ice embryo needs a critical size to grow into a crystal, the area of the nucleating site needs to be





above this critical size. A point defect in a crystal lattice might be too small (e.g. Shevkunov, 2008). If ice nucleation occurred only on a few specific locations, these have to be highly effective and characterized by high nucleation rate coefficients. While observations of deposition nucleation on a crystal may give evidence for preferred locations for ice nucleation, only indirect evidence from refreeze experiments exists for immersion freezing (Vali

et al., 2015).

In deterministic models, active sites are supposed to induce ice nucleation at a characteristic temperature (Vali et al., 2015). The nucleation rate is equal to zero at temperatures higher than the characteristic temperature of the site and equal to infinity beyond that. This implies that no time dependence is involved in nucleation. In a stochastic description, time dependence is introduced by assigning to each nucleation site a characteristic nucleation

rate which is a function of temperature. Ice nucleation as a stochastic process occurring on specific sites can be described by classical nucleation theory (CNT) assuming that heterogeneous nucleation takes place on active site areas, which are often taken as the areas needed by a critical embryo to develop (Marcolli et al., 2007).

When a droplet containing an ice-nucleating particle with an active site is subjected to freezing cycles, the deterministic assumption predicts freezing at exactly the same temperature for every cycle, independent of cooling

rate, whereas the stochastic approach predicts variable freezing temperatures, which depend on the applied cooling rate. The site can be characterized by a nucleation rate, which is a function of temperature and expected to increase with decreasing temperature. When the droplet contains particles with many sites but all of equal quality, the nucleation rate and the rise of the rate with decreasing temperature is higher compared with the case of droplets containing just one nucleation site. In such an idealized case, nucleation rates derived from multiple droplets

are nevertheless characteristic for a specific nucleation site. In most experimental studies such as investigations with continuous flow diffusion chambers (Welti et al., 2012; Lüönd et al., 2010) many particles are investigated and a less steep temperature dependence of heterogeneous nucleation rates compared with the homogeneous case is observed. However, there is strong evidence that the surfaces of most ice-nucleating particles are not uniform with respect to their ability to nucleate ice (e.g. Marcolli et al., 2007; Vali, 2014). Refreeze experiments show that

variations of the freezing temperatures between individual droplets are much smaller than the range covered by freezing experiments with many droplets, in accordance with the assumption that specific sites are responsible for freezing (Vali, 2008; 2014; Wright and Petters, 2013; Peckhaus et al., 2016). If an ice-nucleating sample consists of particles containing sites with different ice nucleation efficiencies, a rate derived from freezing events of many droplets cannot be considered as characteristic of a specific nucleation site type, rather it characterizes a whole

sample with a variety of sites. Moreover, the derived nucleation rate is not purely stochastic, but it has a deterministic component given by the spread of ice nucleation efficiencies of the different sites. If the ice nucleation





ability of a whole sample is wanted, measurements of many droplets are convenient to give a result that is representative for the whole sample. For most natural samples, the sample heterogeneity indeed leads to a large spread of nucleation efficiencies of sites and the temperature dependence is likely to exceed the time dependence (Marcolli et al., 2007; Wright et al., 2013; Wright and Petters, 2013). This was confirmed by a sensitivity study performed by Ervens and Feingold (2012) and is in agreement with Welti et al. (2012), who found the time dependence to be of minor importance for immersion freezing experiments with kaolinite particles. Therefore, a singular or deterministic approach to describe ambient ice-nucleating particles in models may be appropriate and justified. On the other hand, to advance the microphysical understanding of ice nucleation, the presence and properties of ice-nucleating sites need to be investigated in refreeze experiments, where the same sample is subjected to several freezing cycles. Refreeze experiments of a sample containing many different nucleation sites probe only the best one. If a sample is divided into different parts, only one part will contain the particle with the best site, in the other portions less effective sites come into action and will induce freezing at slightly lower temperature. Therefore, in more dilute samples, less efficient sites are probed.

If the slope of nucleation rate increase of single sites is compatible with the one for homogeneous nucleation, a description with CNT is possible by just adjusting one parameter, namely the contact angle. However, when the slope predicted by the parameterization of homogeneous nucleation deviates from the measured one in refreeze experiments a second fit parameter is needed to describe the nucleation rate as a function of temperature. Conceptually, it has been suggested to describe heterogeneous ice nucleation in terms of a static factor, which is specific to the interaction between the nucleating surface and the ice embryo, and a dynamic factor, which accounts for the random timing of the formation of a stable (supercritical) embryo (Vali, 2014). In the present study, we performed refreeze experiments similar to those of Vali (2008) and Wright and Petters (2013) in order to characterize and compare the properties of single nucleation sites. We fitted the freezing rates from the refreeze experiments using three different CNT-based parameterizations from Pruppacher and Klett (1997), Zobrist et al. (2007), and Ickes et al. (2015) under the assumption that ice nucleation occurs on a single site of critical size, namely the most effective one in the sample.

The following samples have been investigated: Hoggar Mountain dust, Arizona test dust (ATD), and birch pollen washing water. Hoggar Mountain dust collected from the Sahara (Pinti et al., 2012) was chosen to represent natural mineral dusts. It is a mixture of minerals originating from a source region of dust aerosols with high shares of clay minerals. A number of field studies have demonstrated the dominant role of mineral dusts to nucleate ice in mixed phase clouds (Sassen et al., 2003; Ansmann et al., 2008; Pratt et al., 2009; Choi et al., 2010; Creamean et al., 2013), and possibly also in cirrus clouds (Cziczo et al., 2013). Arizona test dust (ATD) is a commercial dust





sample that has been used by many groups as a proxy of natural atmospheric mineral dust (Murray et al., 2012). It is a mixture of minerals with a considerable share of microcline, a K-feldspar with a high ice nucleation ability as demonstrated in laboratory experiments (Atkinson et al., 2013; Zolles et al., 2015; Harrison et al., 2016). Pollen is among the primary biological aerosol particles that nucleate ice (Hader et al., 2014). Its importance for

precipitation on the regional scale has been suggested in a number of studies (Pöschl et al., 2010; Prenni et al., 2013; Huffman et al., 2013; Hader et al., 2014). Birch pollen are one of the most efficient pollen species at nucleating ice as high as 264 K (Diehl et al. 2001, 2002; von Blohn et al., 2005; Pummer et al., 2012; Augustin et al., 2013). Pummer et al. (2012) have shown that macromolecules on or within pollen grains are responsible for the ice nucleation activity. These macromolecules are extracted by suspending the pollen grains in water and may be

dispersed in the atmosphere during wetting and drying cycles (Pummer et al., 2012; Hader et al., 2014) and also transported to high altitudes. Birch pollen washing water containing macromolecules with 100 – 300 kDa show similar freezing temperatures as the whole pollen grains (Pummer et al., 2012). Zobrist et al. (2007) performed refreeze experiments of water droplets coated by a nonadecanol monolayer, which arranges in a 2D crystalline lattice on the water surface. The structural match of this 2D crystal with the ice lattice has been considered as key

reason for the good ice nucleation ability of long-chain alcohol monolayers (Popovitz-Biro et al., 1994; Majewski et al., 1995; Knopf and Forrester, 2011). The refreeze experiments by Zobrist et al. (2007) are re-evaluated here assuming that instead of the whole monolayer only an active site in it is responsible for ice nucleation.

The refreeze experiments are analyzed to tackle the following questions: (i) is there experimental evidence that freezing starts from a nucleation site rather than occurring on a random location of an extended ice-nucleating

surface? (ii) Is freezing initiated by always the same nucleation site for each run of a refreeze experiment? (iii) Are nucleation sites stable over the course of a refreeze experiment? (iv) Is one fit parameter enough to describe the properties of an active site or are two fit parameters needed? (v) What is the critical size of an ice-nucleating site? Moreover, the results are set in relation to the microphysical properties of the samples.

## 2 Classical nucleation theory

CNT formulates the Gibbs free energy to nucleate a solid phase from the liquid as the sum of a volume term accounting for the energy released when a molecule is incorporated from the liquid into the solid phase and a surface term accounting for the energy needed to establish the interface between the solid and the liquid phases. The critical size of the embryo is reached when the probability of growth becomes equal to the probability of decay (Vali et al., 2015). Nucleation is described as an activated process with an Arrhenius-type equation, which yields





nucleation rates as a function of the activation energy needed to form the critical embryo (e.g. Fletcher, 1958; Thomson et al. 2015):

$$\omega(T) = A \exp\left(-\frac{\Delta G(T)}{kT}\right)$$  (1)

with the pre-exponential factor $A$ and the activation energy $\Delta G(T)$; $k$ is the Boltzmann constant, $T$ is the absolute

temperature. For first order reactions $A$ has units of $s^{-1}$ and is considered as attempt frequency of a reaction. When the prefactor $A$ is low, the number of nucleation attempts is low and an activated process may not be immediate even if $kT$ is large enough to overcome the energy barrier.

In the framework of CNT, the freezing due to homogeneous nucleation is described by a nucleation rate coefficient given as (Pruppacher and Klett, 1997):

$$j_{\text{hom}}(T) = Z \frac{kT}{h} \exp\left(-\frac{\Delta F_{\text{diff}}(T)}{kT}\right) n_v \exp\left(-\frac{\Delta G(T)}{kT}\right)$$  (2)

where $h$ is the Planck constant, $\Delta F_{\text{diff}}(T)$ is the diffusion activation energy, $n_v$ is the number density of water molecules and $Z$ is the Zeldovich factor described by Zeldovich (1942), which is usually set to 1. $\Delta G(T)$ is the Gibbs free energy described as

$$\Delta G(T) = \frac{16\pi(\sigma(T))^3(V(T))^2}{3[kT\ln(S(T))]^2}$$  (3)

where $\sigma(T)$ is the interfacial energy between ice and the surrounding medium, $V(T)$ is the volume of a water molecule in ice, and $S(T)$ is the ice saturation ratio. The critical embryo radius can be calculated as

$$r_c = \frac{2\sigma(T)V(T)}{kTln(S(T))} \ .$$  (4)

The critical embryo radius calculated with CNT can be validated by testing its consistency with the melting and freezing point depression of ice observed in pores of mesoporous silica (Marcolli, 2014). When pores are too

narrow to incorporate an ice crystal of critical size, ice will not nucleate. Subcritical ice clusters may be produced at a high rate but are inhibited to reach critical size by the confinement in the pores. Marcolli (2014) showed that the CNT-based parameterization by Zobrist et al. (2007) is able to describe the observed melting point depression in pores as a function of temperature and should therefore be well suited to estimate critical embryo sizes. The critical embryo volume for homogeneous ice nucleation predicted by this parameterization is 109 nm$^3$ at 254 K

and increases to 1441 nm$^3$ at 265 K. We therefore consider the energy barrier and critical embryo size predicted by CNT as a quantity with a physical basis.





CNT assumes that heterogeneous freezing occurs on ice-nucleating surfaces that are able to reduce the interfacial energy between the ice embryo and the surrounding. If the critical embryo forms on such a surface, the energy needed to establish the interface is reduced. This leads to a decrease of the energy barrier to form a critical ice embryo. This reduction is described by the contact angle $\alpha$ between the ice phase immersed in liquid water and the heterogeneous surface. The heterogeneous nucleation rate coefficient describing nucleation in contact with an ice-nucleating surface is given as

$$j_{het}(T) = Z \frac{kT}{h} \exp\left(-\frac{\Delta F_{diff}(T)}{kT}\right) n_s \exp\left(-\frac{\Delta G(T) f_{het}(\alpha)}{kT}\right), \qquad (5)$$

where $n_s$ is the number density of water molecules at the ice embryo/water interface. $f_{het}(\alpha)$ describes the change of the Gibbs free energy dependent on the contact angle $\alpha$ due to the influence of ice-nucleating substrates and is described as (Seinfeld and Pandis, 2006)

$$f_{het}(\alpha) = \frac{1}{4}(2 + \cos\alpha)(1 - \cos\alpha)^2. \qquad (6)$$

The volume of the critical embryo reduces to 50% when the contact angle is 90°, and to 33% for a contact angle of 45°.

Several parameterizations of $j_{hom}(T)$ have been proposed in the literature. Three different parameterizations are considered in this study to evaluate the measured data, namely, the parameterization provided by Zobrist et al. (2007), hereafter referred to as Z07, the parameterization given by Pruppacher and Klett (1997), P&K97, and the parameterization from Ickes et al. (2015), Ick15. Differences between these parameterizations are discussed by Ickes et al. (2015) and concern mainly the treatment of $\Delta F_{diff}(T)$, $\sigma(T)$, and $n_s$. P&K97 fitted $\Delta F_{diff}(T)$ from laboratory data and estimated the interfacial energy. They assumed $n_s = 5.85 \cdot 10^{18}$ m$^{-2}$. Z07 parameterized $\Delta F_{diff}(T)$ with measurements from Smith and Kay (1999). The interfacial energy was used as a fit parameter to bring CNT in accordance with homogeneous freezing experiments. They used $n_s = 10^{19}$ m$^{-2}$. Ickes et al. (2015) took $\Delta F_{diff}(T)$ from Zobrist et al. (2007) and the interfacial energy from Reinhardt and Doye (2013). They also used $n_s = 10^{19}$ m$^{-2}$.

If heterogeneous ice nucleation is described by a CNT-based formulation with a reduced energy barrier for critical embryo formation given by $\Delta G(T) f_{het}(\alpha)$, the increase of the heterogeneous ice nucleation coefficient $j_{het}(T)$ with decreasing temperature is tied to the corresponding homogeneous expression ($j_{hom}(T)$) with no possibility for an independent variation of the slope. In this study, we therefore introduce an additional dimensionless prefactor $\beta$ as fit parameter to bring the temperature dependence of CNT-based nucleation rates in agreement with the measured freezing rate increase:



$$j_{\text{het}}(T) = \beta Z \frac{kT}{h} \exp\left(-\frac{\Delta F_{\text{diff}}(T)}{kT}\right) n_s \exp\left(-\frac{\Delta G(T) f_{\text{het}}(\alpha)}{kT}\right). \tag{7}$$

We assume the prefactor $\beta$ to be independent of temperature in the fitted temperature range. A prefactor $\beta < 1$ is needed in case of a shallow slope of the freezing rate increase with decreasing temperature and implies a lower number of nucleation attempts for heterogeneous ice nucleation than predicted from $j_{hom}(T)$, so that even when the

area of the site is large enough to accommodate an ice embryo of critical size, nucleation is not immediate. When $\beta > 1$, the number of nucleation attempts is increased compared with the prediction based on $j_{hom}(T)$ and nucleation is supposed to occur virtually as soon as the temperature is low enough to accommodate a critical ice embryo on the site.

We fit the measured data in two ways: in version V1, we use $\alpha$ as the only fit parameter and $\beta$ is set to unity ($\beta =$

1); and in version V2, we use both $\alpha$ and $\beta$ as fit parameters.

Fits were performed directly to the nucleation rate $\omega(T)$ assuming that nucleation occurs on active sites of critical size:

$$\omega(T) = A_{\text{crit,het}} j_{het}(T) = A_{crit,het} \beta Z \frac{kT}{h} \exp\left(-\frac{\Delta F_{\text{diff}}(T)}{kT}\right) n_s \exp\left(-\frac{\Delta G(T) f_{\text{het}}(\alpha)}{kT}\right). \tag{8}$$

$$A_{\text{crit,het}} = \pi r_c^2 \sin^2 \alpha \tag{9}$$

describes the contact area of an ice embryo with the contact angle $\alpha$ evaluated at the mean freezing temperature of a refreeze experiment, and $r_c$ is the radius of a critical embryo for homogeneous freezing given in Eq. (4). Critical site areas needed to accommodate an ice embryo of critical size, $A_{\text{crit,het}}$, are obtained as a result of the CNT fits to the experimentally determined freezing rates using Eq. (8).

## 3 Statistical description of the ice nucleation process

The statistical evaluation of bulk measurements follows the procedure described in Koop et al. (1997). Here we summarize some of the key aspects of this probability-based description. Ice nucleation is considered to be a stochastic process. This can then be described in terms of a binomial distribution, which provides the probability

$$P_k(m) = \binom{m}{k} p^k (1-p)^{m-k} \tag{10}$$

to observe $k$ nucleation events with the probability $p$ for $m$ attempts to build a critical nucleus (or embryo). The

variance $v$ can be calculated by the formula

$$v = mp(1-p). \tag{11}$$





Since each water molecule in a bulk sample can become the center of a critical nucleus, $m$ can be considered as the number of water molecules in the bulk sample, yielding an $m$ value for our bulk measurements with droplet volumes of about 2 mm$^3$ of ca. $10^{19}$. Due to this large value, Stirling's approximation

$$k! \approx \sqrt{2\alpha k} \; k^k \; e^{-k} \quad \text{if} \quad m - k \gg 1 \quad \text{and} \quad p \ll 1 \tag{12}$$

can be applied and we obtain the Poisson distribution

$$P_k(m) \approx \frac{(mp)^k}{k!} e^{-mp} . \tag{13}$$

The nucleation rate for a single molecule can be written as $p/t$ and for the whole sample the nucleation rate becomes $\omega \equiv mp/t$ (in s$^{-1}$). The Poisson distribution, given in this formulation as

$$P_k(t) = \frac{(\omega t)^k}{k!} e^{-\omega t} , \tag{14}$$

is a function of time and describes the probability to observe in the time interval [0,t] exactly $k$ incidences of nucleation. The probability for zero ($k = 0$) incidences of nucleation, i.e. no freezing at all, is

$$P_0(t) = e^{-\omega t} . \tag{15}$$

Since there is only one incidence of nucleation needed to freeze a bulk sample, freezing iterations have to be performed to obtain a statistically relevant result.

We now consider a small temperature interval (a few tenths of a degree Kelvin), and within this interval, $p$ is assumed to be constant. In the refreeze experiments, the sample passes this interval $n_{tot}$ times (with constant cooling rate). When the number of passes with no nucleation is defined as $n_{liq}(t)$ the probability for no nucleation becomes

$$P_0(t) = e^{-\omega t} \approx \frac{n_{liq}(t)}{n_{tot}} . \tag{16}$$

If we assume that $n_{nuc}$ samples nucleate after times $t_{nuc,i}$ (with $i = 0, 1, ..., n_{nuc}$) and $n_{liq}$ samples stay liquid over times $t_{liq,i}$ (with $i = 0, 1, ... n_{liq}$), the total time $t_{tot}$ is defined by

$$t_{tot} = \sum_{i=0}^{n_{liq}} t_{liq,i} + \sum_{i=0}^{n_{nuc}} t_{nuc,i} . \tag{17}$$

By means of the relation

$$n_{nuc} = \sum_{k=0}^{\infty} k P_k(t_{tot}) = \sum_{k=1}^{\infty} k \frac{(\omega t_{tot})^k}{k!} e^{-\omega t_{tot}} = \omega t_{tot} \sum_{k=1}^{\infty} \frac{(\omega t_{tot})^{k-1}}{(k-1)!} e^{-\omega t_{tot}} =$$

$$\omega t_{tot} \sum_{k'=0}^{\infty} \frac{(\omega t_{tot})^{k'}}{(k')!} e^{-\omega t_{tot}} = \omega t_{tot} \sum_{k'=1}^{\infty} P_{k'}(t_{tot}) = \omega t_{tot} \tag{18}$$





the nucleation rate $\omega$ is obtained as

$$\omega = \frac{n_{\text{nuc}}}{t_{\text{tot}}} . \tag{19}$$

Here, $n_{\text{nuc}}$ is the total number of freezing events observed within the considered time $t_{\text{tot}}$. The nucleation rate coefficient

$$j_{\text{hom}} = \frac{\omega}{V_{\text{sample}}} \quad \text{or} \quad j_{\text{het}} = \frac{\omega}{S_{\text{IN}}} \tag{20}$$

is calculated by dividing the nucleation rate by the sample volume $V_{\text{sample}}$ for homogeneous ice nucleation or by the ice-nucleating surface $S_{\text{IN}}$ for heterogeneous ice nucleation.

The freezing rate $\omega$ can be calculated using Eq. (19), and the uncertainty of freezing rates was calculated following Poisson statistics on the 95 % level. The 95% confidence level $x$ for these measurements assuming Poisson distribution can be calculated as described by Koop et al. (1997). The lower confidence limit, $\omega_{\text{low}}$, is defined such that less than $n_{\text{nuc}}$ nucleation events would occur with a probability $x$ if $\omega_{\text{low}}$ were the true nucleation rate:

$$x = e^{-\omega_{\text{low}}t_{\text{tot}}} \sum_{k=0}^{n_{\text{nuc}}-1} \frac{(\omega_{\text{low}}t_{\text{tot}})^k}{k!} . \tag{21}$$

Correspondingly, for the upper confidence limit, $\omega_{\text{up}}$:

$$x = 1 - e^{-\omega_{\text{up}}t_{\text{tot}}} \sum_{k=0}^{n_{\text{nuc}}} \frac{(\omega_{\text{up}}t_{\text{tot}})^k}{k!} , \tag{22}$$

where $\omega_{\text{low}}$ ($\omega_{\text{up}}$) is the lower (upper) confidence limit for the nucleation rate, $n_{\text{nuc}}$ is the number of observed freezing events within the considered time $t_{\text{tot}}$, and $k$ is the number of nucleation events within $t_{\text{tot}}$ (Koop et al., 1997).

## 4 Experimental setup and procedures

### 4.1 Treatment of samples

Coarse particles were removed from the Hoggar Mountain dust sample by sieving with a 32 μm grid prior to use. No pretreatment was applied to the ATD sample. The mass concentration of Hoggar Mountain dust aqueous suspensions was 0.5 or 5 wt%. The mass concentration of ATD was 5wt%.

The birch pollen washing water was provided by Bernhard Pummer and is from the same birch pollen batch as described in Pummer et al. (2012). The concentration of the birch pollen suspension was 50 mg/ml. Filtration of this suspension was reported to give a 2.4 wt% birch pollen washing aqueous solution (Pummer et al., 2012). The birch pollen washing water was further diluted with water (Molecular Biology Reagent water from Sigma-Aldrich) to obtain mass concentrations with respect to birch pollen grains between 0.001 and 50 mg/ml.



## 4.2 Differential scanning calorimetry

Experiments were conducted with a Differential Scanning Calorimeter (DSC) Q10 from TA Instruments. Re-freeze experiments were carried out by placing about 1.8 – 2 mg (volumes of 1.8 – 2.0 mm$^3$) of the respective suspension into an aluminum pan, covering the drop with mineral oil to avoid evaporation and condensation and closing the pan hermetically. "Water Molecular Biology Reagent" from Sigma-Aldrich was used to prepare the suspensions since it proved to have a lower average freezing temperature compared with our Milli-Q water. The same sample was subjected to repeated freezing runs with 10 K/min and 1 K/min cooling rates. For Hoggar Mountain dust also measurements at constant temperature were performed by cooling the sample to the target temperature and keeping it there for one hour. For every refreeze experiment we took a fresh sample from our stock solution.

For the emulsion measurements 5wt% lanolin was mixed with 95 wt% mineral oil. 80 vol% of this mixture and 20 vol% of aqueous suspension were vigorously stirred to obtain an emulsion as described by Pinti et al. (2012). This suspension was subjected to repeated freezing cycles. The first and third freezing cycles were conducted with a cooling rate of 10 K/min to check the stability of the sample. The second cycle was performed with 1 K/min and was used for evaluation.

For the refreeze experiments, the onset of the freezing peak was evaluated. The evaluation was done using the implemented software "TA Universal Analysis" of the instrument. The DSC is able to detect and control by means of a thermocouple tiny temperature differences between an empty reference pan and the sample pan, which contains the sample of interest. Due to the latent heat release during a freezing event, the resulting temperature difference leads to a heat flux and to a signal in the counteracting electric current applied to the thermocouple. The precision of the DSC temperature measurement is nominally 0.01 K. Depending on the cooling rate, heat transfer limitations result in a temperature gradient within the droplet and the temperature measured at the bottom of the DSC pan does not correspond exactly with the temperature inside the droplet. Thus, for such measurements additional uncertainties have to be considered. To estimate these uncertainties it was assumed that the cooling or heating of the droplet is fully controlled by the contact to the bottom of the pan and that the surrounding air has a negligible influence. The temperature gradient inside the droplet can then be estimated by the thermal conductivity of water. For bulk measurements with 1.8 mm$^3$ droplets and the assumption that the droplet is a semi-sphere the temperature gradient is about 0.7 K for a 10 K/min cooling rate and 0.07 K for a 1 K/min cooling rate.





## 5 Evaluation

### 5.1 Nucleation rates

To calculate freezing rates from the refreeze experiments, the measured freezing temperatures were divided into bins of equal interval width. The interval width was optimized for each dataset subject to the freezing range, the number of freezing events and the resolution of the DSC, which depends on the cooling rate. The Hoggar Mountain dust measurements utilize 0.2 K bins for 1 K/min cooling rate and 0.3 K bins for 10 K/min cooling rate, such that data points were at least distributed between five temperature bins. For birch pollen washing water, freezing temperatures for a sample were spread over a smaller range than for Hoggar Mountain dust, resulting in a bin widths between 0.08 K and 0.2 K. For the evaluation of the water droplets covered with a nonadecanol monolayer the same bin sizes as in Zobrist et al. (2007) were used. Bins were between 0.5 K and 1.5 K in width. At least four temperature bins were populated by freezing events allowing to estimate the temperature dependence of the nucleation rate coefficient. We assume that these observed freezing rates are equivalent to nucleation rates. To calculate nucleation rates $\omega$ (s$^{-1}$) the procedure described by Koop et al. (1997) was applied as summarized in Sect. 3.

### 5.2 Nucleation rate coefficients

#### 5.2.1 Hoggar Mountain dust and ATD

To calculate nucleation rate coefficients $j_{het}$ (cm$^{-2}$s$^{-1}$) from the nucleation rates, two opposing assumptions were applied for Hoggar Mountain dust and ATD, yielding two possible extremes:

(i)   In the conventional manner, a lower limit for $j_{het}$ was obtained by assuming that the whole sample surface is active at nucleating ice. For Hoggar Mountain dust and ATD, the available surface area per sample was calculated based on the mass present in the sample and the BET (Brunauer-Emmett-Teller) surface area, namely 46.3 m$^2$/g for Hoggar Mountain dust (Pinti et al., 2012) and 85 m$^2$/g for the ATD sample (Bedjanian et al., 2013).

(ii)   An upper limit for $j_{het}$ was obtained by assuming only one single site per sample and assuming a critical site size $A_{crit,het}$ calculated at the mean freezing temperature of the experiment with Eq. (8).

#### 5.2.2 Birch pollen washing water

We obtained birch pollen washing water from Pummer et al. (2012), which was prepared by filtration of suspensions of birch pollen grains. The washing water contains macromolecules with an upper limit of 300 kDa mass



corresponding to diameters of 10 nm (assuming spherical shape and a density of 1.5 g/cm$^3$). For the birch pollen washing water three very different assumptions were used to calculate the active surface of the macromolecules:

(i) The lower limit of the nucleation rate coefficient was obtained by assuming that the surfaces of all macromolecules present in the birch pollen washing water contribute to freezing. Based on the information given by Pummer et al. (2012), we calculated the number of macromolecules present in the suspensions assuming that a 50 mg/ml birch pollen suspension yields a 2.4 wt% birch pollen washing water solution consisting of macromolecules of 300 kDa. Assuming spherical shape, the surface of a single macromolecule was calculated and multiplied by the number of macromolecules present in the solution. This yields a value of 10$^{14}$ macromolecules present in a 50 mg/ml pollen bulk droplet. From this, the total surface $A_{tot}$ of all macromolecules was estimated to be about 300 cm$^2$, i.e. some 14 orders of magnitude larger than $A_{crit,het}$.

(ii) The upper limit of the nucleation rate coefficient was obtained by assuming one active site per sample. The area of the active site was taken as $A_{crit,het}$.

(iii) The presence of a homogeneous freezing peak in emulsion measurements of birch pollen washing water reveals that not all macromolecules are active at nucleating ice. Accounting only for the active ones and assuming that all active macromolecules induce freezing at the same temperature leads to an intermediate value for the nucleation rate coefficient. Knowing the droplet size distribution of the emulsions and the size of particles in the pollen washing water, the theoretical homogeneous resp. heterogeneous freezing peak area can be estimated. The probability $P_j$ for a macromolecule to be in a droplet $j$ with a volume $V_j$ is $P_j = V_j/V_{tot}$, where $V_{tot}$ is the total volume of all droplets in the emulsion. Assuming $n$ macromolecules in the emulsion, which are all distributed among the water droplets, the probability for no macromolecule in a droplet $j$ with a volume $V_j$ is $(1-V_j/V_{tot})^n$. The contribution of droplet $j$ to the total heterogeneous and homogeneous peak area $A_{tot}$ is proportional to $V_j/V_{tot}$. The percentage of homogeneous freezing, $p_{hom}$, can then be written as

$$p_{hom} = \sum_{j=1}^{k} \frac{V_j}{V_{tot}} \cdot \left(1 - \frac{V_j}{V_{tot}}\right)^n , \tag{23}$$

where $k$ is the number of droplets. The percentage of heterogeneous freezing $p_{het}$ is given by

$$p_{het} = 1 - p_{hom} . \tag{24}$$

Comparing the theoretical and measured values gives an estimate for the fraction of birch pollen macromolecules that are active. This fraction is about 10$^{-7}$. Estimates based on bulk measurements with different dilutions (see Fig. 2) lead to a similar result. For these bulk measurements the washing water was diluted to 5×10$^{-6}$ mg/ml until freezing occurred at temperatures, at which also pure water may start to freeze (indicated by the horizontal line in Fig. 2). Assuming that at these concentrations hardly any ice-nucleating particles are left in a bulk sample, an ice



nucleation active fraction of about $10^{-7}$ particles was obtained again. Therefore, the number of active macromolecules can be estimated by dividing the total number of macromolecules in a bulk sample ($10^{14}$) by $10^{7}$, yielding a value of $10^{7}$ ice nucleation active macromolecules present in a 50 mg/ml pollen suspension.

### 5.2.3 Nonadecanol coated droplets

For nonadecanol coated droplets, the assumptions that the whole nonadecanol monolayer was ice nucleation active and the presence of only one active site in the monolayer were used to convert from nucleation rate to nucleation rate coefficient.

### 5.4 Spearman's rank correlation coefficient

The Spearman's rank correlation coefficient assesses how well the relationship between two variables can be described using a monotonic function. To evaluate whether a trend is present in the refreeze experiments during repeated freezing cycles, the Spearman's rank correlation coefficient was calculated using time and freezing temperatures as variables. The values are presented in Tables 1 and 2. Numbers around zero indicate hardly any monotonic trend. Numbers close to 1 or -1 indicate a strong monotonic trend.

## 6 Results

### 6.1 Hoggar Mountain dust

Figure 3 shows refreeze experiments performed with Hoggar Mountain dust samples (H1 – H12). With each sample between 21 and 97 freezing cycles were performed. The sequences of freezing temperatures for several samples reveal clear signs of non-stochastic behavior, such as trends or jumps. Therefore, following Zobrist et al. (2007), samples were tested by means of a linear fit for the presence of a trend. When the 95% confidence level of the slope included zero, the samples were considered to be constant in their freezing behavior over the conducted freezing cycles, a necessary (but not sufficient) condition for a stochastic behavior. Samples H1 – H9 performed with 1 K/min cooling rate satisfy the condition of the absence of an overall trend (see Fig. 3), although not all freezing series appear to be purely stochastic. Furthermore, for runs of samples H6 and H9 performed at 10 K/min only a 99.7% confidence level of the slope included zero.

Samples H10 – H12 are examples of refreeze experiments, which show non-stochastic features. This may be a decrease of freezing temperatures by almost 3 K over about eight freezing runs (H10), or an abrupt jump to lower





freezing temperature by almost 2 K from one freezing run to the next (H11), or a slow increase of freezing temperature by 4.5 K over 35 freezing cycles (H12). Such features are clearly non-stochastic and must have been due to modifications (deteriorations or improvements) of the site, which might be due to coagulation, settling, or breakup of aggregates (Emersic et al., 2015).

We calculated Spearman's rank correlation coefficients to check the data concerning a monotonic trend. Table 1 displays the results for Hoggar Mountain dust. The indicated uncertainties represent a 68% confidence interval. Spearman's rank correlation coefficients for Hoggar Mountain dust samples are within ± 0.1 for H3, H4 and H7 indicating hardly any monotonic trend. For H1, H2, H5, H6, H8, H9 and H12 performed at 10 K/min, correlation coefficients are within ± 0.5, indicating at most a weak monotonic trend. Samples H10 and H11 with correlation

coefficients close to 1 show a very strong monotonic trend. Even in the absence of a monotonic trend, refreeze data series do not need to be stochastic. Sample H5 has a weak monotonic trend with a Spearman coefficient of $0.21 \pm 0.09$, but the 4 runs with the highest freezing temperatures all occurred in a row. The probability for this to happen is low ($2.04 \cdot 10^{-4}$). Nevertheless, we cannot exclude that this improbable sequence occurred by chance. We therefore use the samples H1 – H9 for further evaluation.

Panel (a) of Fig. 4 shows the freezing rates for refreeze experiments H1 – H9. While most of the samples freeze at temperatures between 258 and 263 K, samples H4 and H6 freeze at significantly higher temperatures of 263 – 265 K. There are also differences in the slopes of nucleation rates. The steepest increase is observed for the samples H4 and H6, which freeze at the highest temperatures. Freezing rates for H8 and H9 align well for a wide range of cooling rates (10 K/min, 1 K/min and at constant temperature).

Next, we calculate nucleation rate coefficients (in $cm^{-2}s^{-1}$) from the measured freezing rates (in $s^{-1}$). Values of the order of $10^{-6} - 1$ $cm^{-2}s^{-1}$ are obtained when assuming the total surface to be ice-nucleating (panel b), and in the order of $10^{9} - 10^{13}$ $cm^{-2}s^{-1}$ when only one active site per sample is assumed to be responsible for ice nucleation (panel c).

Figure 5 presents fitting results for the refreeze experiment H9 using the three CNT-based parameterizations

Ick15, P&K97 and Z07. Analogous figures for the other samples are displayed in the Appendix Fig. A1. Fit parameters for all evaluated refreeze experiments H1 – H9 are listed in Table 3 using only the contact angle as fit parameter and setting the prefactor $\beta$ to 1 ( version V1), and in Table 4 with prefactor $\beta$ and the contact angle simultaneously fitted (version V2). The tables list also the values of the critical active area $A_{\mathrm{crit,het}}$, which is not a fit parameter but a result of the calculation. For all parameterizations, the Hoggar Mountain dust samples show

slopes much shallower than predicted by CNT when only the contact angle is used as fit parameter (V1). Very good fits can be obtained, when the prefactor $\beta$ is used as second fit parameter (V2). For all Hoggar Mountain



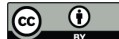

dust samples, the fitted $\beta$ values are $< 1$. For a given parameterization, contact angle and prefactor values show significant differences between samples. The samples H1 and H5 have within statistical variability the same contact angles ($\alpha$(H1) = 36.5 ± 0.7° and $\alpha$(H5) = 36.0 ± 1.7°) and prefactors ($\beta$(H1) = (3.77 − 11.96) × 10$^{-6}$ and $\beta$(H5) = (1.36 − 13.42) × 10$^{-6}$). The same is the case for samples H4 and H6 and samples H8 and H9.  All other samples can be discriminated from one another in terms of contact angles and prefactors. We therefore conclude that Hoggar Mountain dust samples taken from the same suspension show distinctly different behaviors in terms of freezing temperatures and freezing rate increase with decreasing temperature. This supports the assumption that ice nucleation occurs within these samples on nucleation sites that distinctly differ from each other.

## 6.2  Arizona test dust (ATD)

Figure 6 presents the refreeze experiments performed with ATD for 5 wt% suspensions with freezing temperatures in the range of $T$ = 264 – 268 K. The freezing temperature of the first run was always distinctly lower than the subsequent ones. This memory effect ranged from 1 – 4 K.

The sample A5 was evaluated for freezing rates. To evaluate A5 with respect to its stochastic behavior, the first freezing point was omitted. Spearman's rank correlation coefficients for the A5 sample is close to zero. Therefore, this sample can be assumed to be without a monotonic trend, after removing the initial memory effect by omitting the first point.

Figure 7 shows the CNT-based fits to the freezing rates for sample A5 assuming that freezing occurred on a single nucleation site. The increase of freezing rate with decreasing temperature is much shallower than can be fitted with the CNT-based parameterizations, if the contact angle is the only fit parameter (V1).  If contact angle $\alpha$ and prefactor $\beta$ are used simultaneously as fit parameters (V2), very good fits are obtained for all CNT-based parameterizations. The last lines in Tables 3 and 4 show the fit parameters and the critical heterogeneous surface $A_{\mathrm{crit,het}}$ for the different CNT-based parameterizations. Similar to Hoggar dust, the prefactor $\beta$ needs to be very small to reach agreement with the shallow increase of freezing rate with decreasing temperature.

## 6.3  Birch pollen washing water

Figure 8 shows refreeze experiments of birch pollen washing water samples performed with a cooling rate of 1 and 10 K/min. In all experiments, freezing occurred in the temperature range from 254 to 261 K, but individual samples froze over a much narrower temperature range of typically $< 1$ K. The samples were again tested with a linear fit for trends. For samples P1 and P8 the first few runs showed lower freezing temperatures than all subse-





quent ones, which might be due to a memory effect. These first runs were therefore excluded from the test. Samples P1 – P7 satisfy the 95 % confidence level condition, but not so P8 and P9. For samples P6 and P7 refreeze experiments were also performed at a cooling rate of 10 K/min giving distinctly lower freezing temperatures. There is no overlap in freezing temperatures for these two cooling rates.

Spearman's rank correlation coefficients were calculated to check the data concerning a monotonic trend. The results for the birch pollen washing water are shown in Table 2. The uncertainties given in the table represent a 68% confidence interval. Spearman's rank correlation coefficients for birch pollen washing water samples P1-P5, P6 (10 K/min) and P7 (10 K/min) are ±0.2, indicative of a very weak monotonic trend. Correlation coefficients for P6 (1 K/min), P7 (1 K/min), P8 and P9 are significant different from zero.

Panel (a) of Fig. 9 shows freezing rates for the samples P1 – P9. Sample P3 (50 mg/ml pollen) freezes at distinctly higher temperature than all other samples. There is no significant difference in freezing temperatures for 50 mg/ml and 0.1 mg/ml samples. However, the more dilute P6 sample (0.001 mg/ml) freezes at almost 2 K lower temperature than all other ones. Fig. 2 shows the dependence of freezing temperatures on suspension concentration. It can be seen that the average freezing temperature of birch pollen washing water first decreases gradually

from 257 K to 253 K for a dilution from 50 mg/ml to $5 \times 10^{-6}$ mg/ml, and upon further dilution abruptly drops into the range where also pure water bulk samples may freeze as indicated by the black line at T = 252.5 K in Fig. 2.

The slope of freezing rate with temperature is similar for all refreeze experiments irrespective of solution concentration or cooling rate, with the exception of P8 which shows a distinctly stronger freezing rate increase with de-

creasing temperature. However, experiments performed with cooling rates of 10 K/min and 1 K/min do not fall on one line, but occur with similar freezing rates just at ~1 K lower temperature for 10 K/min compared with 1 K/min. This behavior of the birch pollen samples is in clear contrast to the behavior of Hoggar Mountain dust samples, which showed a good alignment of freezing rates acquired with different cooling rates (see Fig. 4). To check whether the misalignment of the 10 K/min and 1 K/min freezing rates of the birch pollen samples is influ-

enced by the very narrow bin intervals (0.15 K), we varied the bin widths for the 10 K/min experiments. The results in Fig. 10 show that freezing rates are independent of the choice of bin widths ($\Delta T = 0.15 - 1$ K). An alternative explanation might be an induction time required for the ice embryo to grow large enough to be detected in the DSC instrument or due to heat transfer limitations in the pan as discussed in Sect. 4.2.

Similar to the derivation of nucleation rate coefficients for the Hoggar Mountain dust samples, we also applied

for the pollen washing water different assumptions to convert freezing rates to freezing rate coefficients, as de-



scribed in detail in Sect. 5.2.2, yielding very different values for $j_{\text{het}}$, as is shown in Panels (b – d) of Fig. 9. Panel (b) shows freezing rate coefficients in the range $10^{-4} – 10^3$ cm$^{-2}$s$^{-1}$, when assuming that the whole birch pollen washing water consists of macromolecules and that the whole surface of all macromolecules is ice-nucleating. With this assumption, the sample P6, when cooled with 1 K/min, has higher nucleation rate coefficients than the

other samples, because it has the lowest concentration and thus the lowest active area. Conversely, assuming only one active site per sample (Fig. 9d), nucleation rate coefficients in the order of $10^9 – 10^{13}$ cm$^{-2}$s$^{-1}$ are obtained. In Fig. 9c we assume that a small fraction of the birch pollen washing water contains active macromolecules. In Sect. 5.2.2 we estimated this fraction to be $10^{-7}$. With this assumption, the resulting nucleation rate coefficients are in the range $10^1 – 10^{10}$ cm$^{-2}$s$^{-1}$.

Figure 11 presents curves fitted to the refreeze experiment P7 for cooling rates of 10 K/min and 1 K/min for the three CNT-based parameterizations Ick15, P&K97 and Z07. For both cooling rates, P7 shows a slightly steeper slope than could be fitted when only the contact angle was used as fit parameter (V1). Analogous figures for the other samples are given in the Appendix Fig. A2. Fit parameters for all evaluated refreeze experiments P1 – P9 are summarized in Tables 5 and 6. When only the contact angle is used as fit parameter, the fitted contact angles

for most experiments are significantly different from each other (Table 5), but the steep increase of freezing rates with decreasing temperature could not be realized for all samples (see Fig. A2). When the contact angle and the prefactor β are used as fit parameters (V2), good agreement is obtained. For most birch pollen washing water samples the fitted β values are > 1 implying a steeper increase of freezing rate with decreasing temperatures than predicted by the CNT-based parameterizations. However, the $\beta$-values are not well constrained by the fit as can

be seen from the large uncertainties associated with them (Table 6). Worth mentioning are sample P4 with the lowest prefactor β = 0.006–0.0855 and sample P8 with a huge prefactor of $(6.2–513) \times 10^{41}$. Fits of version V2 to samples P1, P2, and P5 yield contact angles that are identical within the observed variability, while the other samples can be differentiated from one another based on their $\alpha$ and $\beta$ values. This strongly suggests that at least for some of the birch pollen washing water samples ice nucleation always occurs on the same site, i.e. on the

same macromolecule. The freezing rates of the samples measured with cooling rates of 10 K/min and 1 K/min (P6 and P7) do not coincide, but those measured with 1 K/min freeze at ~1 K higher temperature. Nevertheless fitting the freezing rates with CNT gives within the observed variability the same contact angles for the two cooling rates (see Table 6), in agreement with ice nucleation occurring on the same site for both cooling rates.





## 6.4 Nonadecanol coated droplets

Zobrist et al. (2007) performed refreeze experiments of water droplets coated by a nonadecanol monolayer for droplets with radii between 31 μm and 1100 μm. They calculated nucleation rate coefficients from the freezing rates assuming that the whole surface of the nonadecanol monolayer is nucleating ice and tried to describe the

nucleation rate coefficients as a function of temperature with CNT using the contact angle as fit parameter. They could reconcile their measurements with CNT only by assuming a temperature dependent contact angle. We re-evaluate their freezing rate data for the nonadecanol samples N1 – N6, assuming that single sites were the location of freezing instead of the whole surface. Each sample is therefore fitted separately with Eq. (8). Figure 12 shows the refreeze experiments for droplet N2 with a radius $r = 1100$ μm in panel (a) and for droplet N6 with $r =$

31 μm in panel (b). Analogous figures for experiments N1 ($r = 1100$ μm), N3 ($r = 370$ μm), N4 ($r = 320$ μm), and N5 ($r = 48$ μm) are shown in Fig. A3 in the Appendix. Fit parameters for nonadecanol droplets N1 – N6 are given in Tables 7 (V1) and 8 (V2). The freezing temperature of nonadecanol coated water droplets decreases significantly with decreasing surface area of the droplets. The droplets (N1 and N2) with $r = 1100$ μm freeze at $T_{fr} =$ 260 – 265 K, the droplets with $r = 370$ or 320 μm between $T_{fr} = 256 – 262$ K and the ones with $r = 48$ or 31 μm at

$T_{fr} = 248 – 252$ K. Fits with $\beta = 1$ (V1) show much too steep slopes compared with the measurements for the samples N1 – N4. The samples N5 and N6 show a steeper slope, already reasonably represented by V1. When the prefactor $\beta$ is fitted as well, the fits of the droplets N1 – N4 improve, however, the freezing rates at the highest temperatures are still not reproduced well. Only a few runs populate the bins at higher temperatures, and their freezing rates are associated with large uncertainty ranges. Therefore, they were given less weight for the fits

shown in Figs. 12 and A3. However, when the fitted curves were forced to pass through the lowest and highest data points by increasing their weighting (not shown), the fit quality decreased for the points measured in between since the resulting curves were too bowed. An improved fit could also not be obtained when the whole surface was considered to be ice-nucleating. Table 8 for V2 shows that the contact angle and the prefactor $\beta$ increase with decreasing droplet size. For the smallest droplets (N5, N6) the prefactor $\beta$ is of the order of unity

(0.001 – 1000) and the contact angle is above 50° for all parameterizations. For the largest droplets (N1, N2) the prefactor is around $10^{-7}$ – $10^{-8}$ and the contact angle is below 32° for all parameterizations. Droplets of similar sizes (N1/N2; N3/N4; N5/N6) have similar contact angles and prefactors.



## 6.5 Emulsion measurements

In Fig. 1 typical thermograms for emulsion measurements of Hoggar Mountain dust (panel a), ATD (panel b) and birch pollen washing water (panel c) are shown. The heterogeneous freezing peak of the Hoggar Mountain dust sample is quite broad compared with those of ATD and birch pollen washing water. This difference in width can be explained by the composition of the samples. For ATD, already Marcolli et al. (2007) showed that the observed dependence of the heterogeneous freezing temperatures cannot be described by assuming a constant contact angle for all ATD particles. Rather, the ice-nucleating sites of ATD particles are required to be of different qualities. Larger particles have on average more and better active sites than smaller ones. This behavior comes to fruition even more in the case of Hoggar Mountain dust, which is a mixture of various minerals which are ice-nucleating at quite different temperatures (Pinti et al. 2012). In contrast to the bulk measurements, no memory effect was observed for ATD emulsions. With an onset of 255 K, the heterogeneous freezing peak of the emulsion made from the birch pollen washing water exhibits a clear overlap with the freezing temperatures observed for bulk measurements. This confirms that macromolecules are active in both, bulk and emulsion samples.

## 7 Discussion

### 7.1 Nucleation on active sites

In the following, we investigate the refreeze experiments for evidence against or in favor of ice nucleation at active sites. Sudden jumps of freezing temperature during refreeze experiments are evidence that specific singular features in the samples are the nucleating entity, which might be fragile and can vanish or emerge during the course of a refreeze experiment.

#### 7.1.1 Hoggar Mountain dust

Refreeze experiments of Hoggar Mountain dust showing sequences with trends or even jumps are strong evidence that freezing occurs on particular sites in these samples. For the H11 sample shown in Fig. 3, the freezing temperature first shows a decrease when after about 40 runs it suddenly drops and remains for the rest of the experiment quite constant at a value ~3 K lower than before. Such a drop points to freezing occurring on a single site, which suddenly becomes inactive and from then on freezing occurred on the next best site. Furthermore, freezing temperatures before the drop give the impression that freezing on this site was not fully stochastic. The samples H10 and H12 show less abrupt transitions, which might be related to a site that remained dominant but underwent modifications during the course of the experiment. The samples H1, H2, H5, H6, H8 and H9 show a





weak monotonic trend, which could be due to slight modifications of the ice-nucleating site during the course of the experiment. Nevertheless, nucleation sequences with such trends fulfill the criteria for evaluation with CNT as long as nucleation supposedly occurred always at the same site, even if this site is not completely stable in time.

Hoggar Mountain dust consists of a mixture of minerals with high shares of the clay minerals smectite/montormillonite, illite, and kaolinite, and minor contributions of quartz and the feldspars sanidine and plagioclase (Kaufmann et al., 2016). However, nucleation on the best sites present in bulk samples (Pinti et al., 2012) does not need to be closely related to the prevailing minerals in the sample. It is therefore not clear whether a specific mineral component or rather a non-mineralogical component present in the collected dust is responsible

for ice nucleation. This further supports the interpretation that freezing occurs on distinct sites that are different for different samples. The evaluation with CNT of the refreeze experiments with Hoggar Mountain dust shows that individual samples taken from the same stock solution can be discriminated based on their contact angles and prefactors. This together with the heterogeneity of the sample and the jumps and trends observed for the time sequences of some samples, supports the notion that ice nucleation occurs on specific sites on the sample surface.

However, it is not clear whether these active sites originate from a specific mineral component or even biogenic components in the dust sample (Conen et al., 2011; Tobo et al., 2014; O'Sullivan et al., 2014). Moreover, the activity of sites could be influenced by coagulation or the breakup of aggregates (Emersic et al., 2015).

### 7.1.2 ATD

Only a few refreeze experiments were performed with ATD. For this limited dataset, we did not observe non-

stochastic behavior such as trends or unexpected jumps, but all samples showed a pronounced memory effect. Wright and Petters (2013) performed refreeze experiments with ATD and observed jumps similar to the ones that we observed for the Hoggar Mountain dust sample but they did not mention a memory effect. ATD is a complex mixture of minerals with a considerable share of microcline (20 – 30 %) (Atkinson et al., 2013), which is a K-feldspar with exceptionally high heterogeneous ice nucleation temperatures. Microcline samples showed high

freezing temperatures from $T = 264 – 272$ K in bulk freezing experiments (Kaufmann et al., 2016) similar to the ones performed in this study. Therefore, microcline is most probably the mineral component responsible for freezing of bulk samples. The experiments by Wright and Petters (2013) were performed with smaller droplets (15 – 120 µm diameter) containing only few particles. Freezing occurred at $T = 236 – 253$ K. Microcline will therefore just be one among the various mineral components responsible for freezing in these experiments. The

freezing can therefore not be ascribed to microcline alone in these experiments, in contrast to the experiments





performed in this study. If the memory effect is due to the microcline component, it may explain why Wright and Petters (2013) did not observe it. Zolles et al. (2015) attribute the high ice nucleation activity of K-feldspars to an intrinsic property of the surface. They hypothesize that the surface cations released into the surface bilayer may interact with water to enhance or inhibit ice formation. Also, the ion charge density of the cations of the mineral was suggested to influence ice nucleation. The memory effect might therefore be related to surface characteristics involving the cation distributions, which might change once the surface has been covered with ice. Indeed, the memory effect in our ATD samples is typically confined to the very first run. The limited number of refreeze experiments with ATD performed for this study does not allow for detailed characterization of the ice nucleation activity of microcline. A dedicated study with refreeze experiments performed on pure microcline samples might help to elucidate whether this mineral possesses surfaces with small patches of high ice nucleation probability or larger surface areas with lower but uniform ice nucleation probability.

### 7.1.3 Birch pollen washing water

There is clear evidence from the emulsion measurements that only a small fraction of the birch pollen macromolecules are ice nucleation active. We estimate this fraction to be $10^{-7}$ (see Sect. 5.2.2) based on emulsion freezing experiments and the dilution series shown in Fig. 2. A 50 mg/ml sample should therefore contain $10^7$ active macromolecules while this number reduces to 1 for a $10^{-5}$ mg/ml sample. These numbers are consistent with the freezing experiments of water droplets activated from a birch pollen washing water aerosol performed by Augustin et al. (2013). They observed a frozen fraction of 0.03 for 800 nm particles at 254 K, which translates into an ice-nucleating fraction of macromolecules of $4\times10^{-8}$ assuming that the whole sample consists of macromolecules with 300 kDa. While our 50 mg/ml samples contain a high number of ice-nucleating macromolecules, not all of them induce freezing at the same temperature. The emulsion measurement (Fig. 1c) shows a heterogeneous freezing peak with onset at about 255 K that stretches to below 245 K, and then fades away. Heterogeneous freezing occurring in a temperature range is in agreement with Augustin et al. (2013). They observed the highest freezing temperatures at 254 K with frozen fractions of 0.007 and 0.02 for 500 and 800 nm particles, respectively. The frozen fraction increased, when temperature was lowered, reaching a plateau with no further increase at 245 K. Augustin et al. (2013) further reported results from Pummer et al. (2012), who investigated droplets in the size range from 10 – 200 µm diameter and observed an increase of frozen fraction from $2.5\times10^{-3}$ at 257 K to full activation at 253 K for 50 mg/ml samples. We can therefore assume that only very few macromolecules are active at the highest temperature. This conclusion is supported by the fits of freezing rates obtained from the different CNT-based parameterizations, which yield significantly different contact angles $\alpha$ and prefactors $\beta$ for most





samples. This indicates that at least for some samples ice nucleation likely occurs always on the same macromolecule during the course of a refreeze experiment.

### 7.1.4 Nonadecanol coated droplets

The refreeze experiments of water droplets coated with a nonadecanol monolayer show a clear decrease of freezing temperature with decreasing surface area of the droplets. The 1100 μm radius droplets freeze between 260 and 265 K, the droplets with radii of 370 μm and 320 μm between 256 and 262 K and the ones with radii of 48 μm and 31 μm between 248 and 252 K. Zobrist et al. (2007) evaluated these results within the framework of CNT assuming that the whole surface of the nonadecanol monolayer is ice nucleation active. They obtained best agreement assuming a temperature dependence of the effective contact angle described by the linear function $\alpha(T) = 571.50 - 2.015 \times (T/\text{K})$, yielding contact angles from 37.5° at $T = 265$ K to 71.8° at $T = 248$ K. They explained this temperature dependence by assuming a reduced compatibility of the alcohol monolayer with the ice embryo as the temperature decreases due to the decreasing mobility of the alcohol molecules on the water surface which inhibits rearrangement of the alcohol molecules at the water surface. Vali (2014), on the other hand, speculated that the monolayers formed by long-chain alcohols are not simple, smooth surfaces but may have discontinuities of various kinds such that ice nucleation occurs on specific nucleation sites and not on the whole monolayer surface. In this study, we re-evaluated the freezing rates determined by Zobrist et al. (2007) assuming that freezing occurred on sites of critical size. Fitting the freezing rates separately for the individual refreeze experiments using the contact angle $\alpha$ and prefactor $\beta$ as fit parameters, yielded for the smallest droplets prefactors $\beta$ around 1 and contact angles above 50°, irrespective of the choice of CNT parameterization. For the largest droplets the prefactor is in the order of $10^{-8}$ and the contact angle is below 32°. Droplets of similar size gave contact angles that are identical within the observed variability. This indistinguishability supports the notion that long-chain alcohol monolayers provide an extended surface with a relatively uniform ability to nucleate ice. However, to substantiate this conjecture more refreeze experiments of droplets with the same size would be needed.

### 7.2 Critical site area

In the framework of CNT, freezing only occurs, when the embryos developing on a site can reach the critical size to grow into a crystal. Because the critical embryo size increases with increasing temperature, also the critical size of a nucleating site increases with temperature. In this study, critical site areas needed to accommodate an ice embryo of critical size, $A_{crit,het}$, are obtained as a result of the fits to the experimentally determined freezing rates using Eq. (8). All three CNT-based parameterizations yield critical areas in the same size range. This is an indica-



tion that the determined values are well constrained and might indeed have a physical basis. Critical site areas, calculated with the three CNT-based parameterizations are $A_{crit,het} = 16 - 39$ nm$^2$ for Hoggar Mountain dust with freezing temperatures $T_{fr} = 258 - 265$ K. For the ATD sample with $T_{fr} = 267 - 268$ K the critical embryo size ranges from $A_{crit,het} = 39 - 52$ nm$^2$ for the different CNT-based parameterizations. Birch pollen washing water

samples freeze in the range $T_{fr} = 254 - 261$ K with $A_{crit,het} = 20 - 50$ nm$^2$. Finally, $A_{crit,het}$ for the nonadecanol samples decrease from $16.1 - 27.2$ nm$^2$ for the $r = 1100$ µm droplets with $T_{fr} = 260 - 265$ K, to $13.2 - 21.6$ nm$^2$ for the $r = 370/320$ µm droplets with $T_{fr} = 256 - 262$ K and finally to $A_{crit,het} = 10.4 - 16.1$ nm$^2$ for the $r = 48/31$ µm droplets with $T_{fr} = 248 - 252$ K. These critical site areas show a temperature dependence and are larger at higher temperatures. They are in the same size range as the ice nucleation active area of proteins expressed by the

bacteria *Pseudomonas syringae (P. Syringae)* and *Erwinia herbicola*, which are active at $263 - 265$ K and have a mass of 150 kDa (Yankofsky et al., 1981; Govindarajan and Lindow, 1988; Budke and Koop, 2015; Pandey et al., 2016). Kajava and Lindow (1993) determined the area of the minimum ice-nucleating site of *P. Syringae* as 25 nm · 2.5 nm $= 62.5$ nm$^2$, corresponding to the area on the protein that shows a lattice match with ice. Critical nucleus surface areas, $A_{crit,het}$, estimated in this study are in general agreement with this number.

**7.3 Fit parameters α and β**

In this study, we fitted the observed freezing rates of refreeze experiments using three different CNT-based parameterizations (Ick15, P&K97 and Z07) together with the assumption that freezing occurs on single sites of critical size at the mean freezing temperature of the refreeze experiment. The different parameterizations gave slightly different values of contact angles, prefactors, and $A_{crit,het}$, but were very similar in their ability to fit the

data. When the contact angle was used as only fit parameter (V1), the parameterizations underrated or overrated the increase of freezing rate with decreasing temperature depending on the sample. If the contact angle $\alpha$ and the prefactor $\beta$ were used as fit parameters (V2), good fits could be obtained for most refreeze experiments. This shows that ice-nucleating sites need to be characterized by two parameters. While the $\alpha$ parameter describes the reduction of the energy barrier in the presence of an ice-nucleating surface, the interpretation of the prefactor $\beta$ is

less obvious. There are different explanations conceivable for the need of a prefactor $\beta$ as additional fit parameter:

(i)   If some sites were not constant in quality from one freezing cycle to the next, ice nucleation on such sites would not be fully stochastic. In this case, it would not be correct to describe the freezing temperature sequences with a constant contact angle $\alpha$. When variability of $\alpha$ is mistaken as random fluctuations of freez-



ing temperatures, a low value of prefactor $\beta$ would be fitted. The presence of a monotonic trend or an improbable sequence of freezing temperatures are indications that nucleation indeed was not fully stochastic. However, there is no criterion available to discriminate stochastic variations of freezing temperature from variations due to variability of $\alpha$.

5  (ii)  A high number of sites active at the same temperature instead of only one or a few would result in a prefactor $\beta > 1$ because each site would contribute to the total frequency of nucleation attempts.

(iii)  For homogeneous ice nucleation the kinetic prefactor is considered to account for the rate at which water molecules are transferred into an ice germ (e.g. Ickes et al., 2015). If the presence of a surface changed this rate because it e.g. influences the orientation of water molecules, the additional fit parameter $\beta$ could account for this. A prefactor $\beta < 1$ would describe an unfavorable orientation of water molecules for the transfer into the growing ice embryo leading to a reduced number of successful nucleation attempts. A prefactor $\beta > 1$, on the other hand, would mean a favorable orientation of water molecules for incorporation into the ice embryo leading to an accelerated nucleation process.

(iv)  If different orientations of water molecules on a surface were energetically similar but only one of them were suited to develop into an ice embryo, nucleation could only occur at times when this favorable arrangement is realized. This would correspond to a reduction of the number of nucleation attempts compared to a case when one preferred orientation of water molecules exists on a surface that promotes ice nucleation. In such a case, $\beta$ would be $< 1$.

(v)  Kinks, cracks or screw dislocations next to a site could orient water molecules favorably to develop critical ice embryos on a site. This would increase $\beta$ compared with the case of a site on a flat surface.

In case of explanations (i) and (ii), the prefactor $\beta$ is just a correction factor lacking a fundamental physical meaning, but accounts for inadequacies of the conjectures for the fit, namely for the assumptions of an ice nucleation active area of critical size (point i), and for the assumption of constant $\alpha$ during the course of the experiment (point ii). Explanations (iii) – (v) imply that the number of nucleation attempts can be lower or higher than predicted by $j_{hom}(T)$ and should be considered as a characteristic of a nucleation site. In the following, we will relate the fit parameters $\alpha$ and $\beta$ to the specific properties of the investigated ice nuclei.

### 7.3.1  Hoggar Mountain dust and ATD

For Hoggar Mountain dust and ATD, the prefactor $\beta$ is low ($10^{-2} – 10^{-9}$). There might be a low bias of $\beta$ if variability of $\alpha$ is taken as random fluctuations of freezing temperatures (point i). Nevertheless, this is likely to be of





minor effect, because there is no correlation evident between monotonic trends of the time series and $\beta$-values. A low value of the prefactor $\beta$ indicates that the ice-nucleating surface is not effective at growing ice embryos of critical size. Even if the temperature has dropped low enough to overcome the energy barrier to form a critical ice embryo on the nucleation sites of Hoggar Mountain dust and ATD, embryos of critical size might form only in-frequently. Pedevilla et al. (2016) investigated the most easily cleaved (001) surface of microcline with ab initio density functional calculations. They demonstrated that water does not form ice-like overlayers in the contact layer, however, they identified contact layer structures of water that induce ice-like ordering in the second over-layer. If these structures are only one among several water structures and develop only infrequently, this might explain a low frequency of freezing attempts, i.e. $\beta < 1$.

### 7.3.2 Birch pollen washing water

For the macromolecules present in the birch pollen washing water the prefactor $\beta$ ranges from 1 – 10 000 for most samples at $T_{fr} = 256 - 261$ K and increases even to ~$10^{40}$ for P8 which freezes at $T_{fr} = 258.5 - 259$ K. This indicates that nucleation attempts are very frequent and the sample freezes immediately once the temperature has dropped low enough to overcome the energy barrier for critical embryo formation. The high values of $\beta$ might indicate that many macromolecules induce freezing at similar temperatures so that they alternate in inducing ice nucleation from run to run and thus increase the effective surface on which ice nucleation may take place (point ii). However, given the variability of freezing temperatures between samples and the low number of macromole-cules that are ice-nucleating at the highest observed freezing temperatures, it is unlikely that the number of mac-romolecules that contribute to the active surface is high.

Assuming sizes of the birch pollen macromolecules of 100 – 300 kDa as inferred by Pummer et al. (2012), the surface area should range from 111 nm$^2$ to 232 nm$^2$ assuming a density of 1.5 g/cm$^3$ for the macromolecules. If we compare this area with the range of calculated critical site area of 20 – 50 nm$^2$, a considerable part of the mac-romolecules' surfaces should be involved in ice nucleation. The macromolecules in birch pollen washing water are considered to be polysaccharides (Pummer et al., 2012, 2013, 2015; Augustin et al., 2013). Pummer et al. (2015) attribute the ice nucleation ability to a hydration shell around the polysaccharides. This hydration shell is considered to form an ice template that does not randomly dissociate like ice embryos in homogeneous ice nucle-ation and should therefore be a perfect ice nucleator. Such a stable shell might indeed be a reason for the high $\beta$ values.



### 7.3.3 Nonadecanol coated droplets

For the larger droplets with radii $r = 1100$ µm and $r = 370/320$ µm covered with the nonadecanol monolayer, the prefactor $\beta$ is small ($10^{-6} - 10^{-8}$), but for the small droplets with $r = 48/31$ µm it is quite large ($10^{-3} - 10^{2}$). The measured slope of freezing rate increase with decreasing temperature was even flatter than could be fitted with a

prefactor $\beta = 1$. It can be seen in Fig. 1 of Zobrist et al. (2007) that all experiments have random outliers to higher temperatures which populate the highest temperature bins. This would mean that at high temperatures, the freezing is limited by the frequency of nucleation attempts because the surface does not offer features that facilitate the aggregation of water molecules into ice-like subcritical clusters that eventually grow to critical size. Investigating $C_{31}H_{63}OH$ alcohol monolayers, which induce freezing at about 271 K, by grazing incidence X-ray diffraction

showed that the coherence length between the monolayer and the ice lattice was only ~2.5 nm corresponding to about five lattice spacings and was rationalized by assuming multiple ice nucleation sites separated on average by about $5 - 6$ nm (Popovitz-Biro et al., 1994; Majewski et al., 1995). A close match between the ice lattice and the monolayer only extends 3 nm in $a$ and 5 nm in $b$ direction. These values yield critical site areas in the same range as the ones calculated for the nonadecanol monolayers from the CNT-based fits (see Table 8). The spacing of the

2D lattice of the nanodecanol monolayer might be temperature dependent such that the lattice fit between the monolayer and ice deteriorates with decreasing temperature. The memory effect observed for this sample is discussed as structural rearrangement within the alcohol monolayer (Seeley and Seidler, 2001; Zobrist et al., 2007). The interaction between the lattice of ice and the 2D crystalline monolayer might lead to a rearrangement of the long-chain alcohols into a structure with improved lattice match and enhanced ice nucleation efficiency. This

supports the interpretation given in Zobrist et al. (2007) that the formation of a critical embryo is favored by lower temperatures and the molecular rearrangement is favored by higher temperatures because the flexibility of the monolayer to adapt to the ice structure decreases with decreasing temperature.

### 8 Summary and conclusions

This study presents freezing rates determined from refreeze experiments using Hoggar Mountain dust, Arizona

Test Dust (ATD) and birch pollen washing water as heterogeneous ice nuclei. These samples were analyzed using three parameterizations of CNT. Additionally, nonadecanol refreeze experiments from Zobrist et al. (2007) were re-evaluated. The presented analysis leads to the following microphysical insights:

- *Presence of preferred nucleation sites:* For Hoggar Mountain dust, ATD and the pollen washing water, there were significant differences in freezing temperatures between samples taken from the same stock





solution. Such differences are not compatible with the assumption that ice nucleation occurs at a random location of a large uniform surface. The experimental basis for the nonadecanol monolayers was too small to come to the same conclusion. Six time sequences of refreeze experiments from droplets of different size were analyzed. Droplets of the same radius were indistinguishable from each other with respect to their freezing temperatures. This is compatible with the assumption that freezing takes place at a random location on a large surface.

– *Stability of sites and randomness of nucleation:* While some of the time sequences observed for Hoggar mountain dust, ATD, and birch pollen washing water were in accordance with stochastic freezing, others showed jumps and trends in the sequence of freezing temperatures indicating that some sites were not stable during the course of the experiment. This is in accordance with Vali (2014) and Wright and Petters (2013) who also evidenced limitations of the stability of sites.

– *Description with CNT:* For the analysis of the experimental data with CNT it was assumed that always the same site is responsible for freezing and that this site is stable and of critical size. Three CNT-based parameterizations were used to describe freezing rates as a function of temperature. All of them led to similar results. For Hoggar Mountain dust, ATD and larger nonadecanol coated water droplets the experimentally determined increase of freezing rate with decreasing temperature is shallower than can be described by CNT using the contact angle as only fit parameter. The opposite is true for birch pollen washing water and small nonadecanol coated water droplets: the observed increase of freezing rate is steeper than can be fitted by CNT-based parameterization relying on the contact angle as only fit parameter. Good agreement of observations and calculations for most experiments were obtained when a prefactor $\beta$ was introduced as second fit parameter.

– *Critical site size:* the description of heterogeneous nucleation with CNT implies that nucleation occurs on sites with a minimum (critical) surface area so that embryos that develop on them can reach the critical size to grow into ice crystals. CNT provides an estimate of the size that is needed to accommodate the critical embryo. This size is in the range of $10 - 50$ nm$^2$ for the investigated ice nuclei. The required size decreases with decreasing nucleation temperature. Sizes in this order of magnitude are in agreement with the area of the minimum ice-nucleating site that was determined for *P. Syringae*. We therefore suggest that ice-nucleating surfaces have to be searched for features in this size range to identify ice-nucleating sites.

– *Interpretation of fit parameters:* The energy barrier of nucleation is reduced when the ice embryo forms at an ice-nucleating surface. The reduction of Gibbs energy is described by the contact angle $\alpha$, which





was used in this study as first fit parameter. To adjust the slope of freezing rate increase with decreasing temperature predicted by the three CNT-based parameterizations to the measured one, a second fit parameter in the form of a prefactor $\beta$ was needed. If the assumption of nucleating area of critical size and constant $\alpha$ is valid, the prefactor $\beta$ modifies the frequency of nucleation attempts predicted by CNT. If $\beta$

5 > 1, there are many nucleation attempts and nucleation occurs immediately when the temperature is low enough so that the active site area is large enough to accommodate a critical embryo. This is the case for the birch pollen washing water and the small droplets coated with nanodecanol. If $\beta < 1$, the number of nucleation attempts is low and the increase of freezing rate with decreasing temperature is shallow. This is the case for Hoggar Mountain dust, ATD, and the large droplets coated with nonadecanol.

**Appendix A**

The figures in this appendix present fitting results for the three CNT-based parameterizations Ick15, P&K97 and Z07 to refreeze experiments of Hoggar Mountain dust containing water droplets (Fig. A1), birch pollen washing water droplets (Fig. A2) and nonadecanol coated droplets (Fig. A3). The fitting results for the samples presented

15 in this appendix are in accordance with the fitting results for samples presented in the main part of this publication.

**Acknowledgments**

This work was supported by the Swiss National Foundation, project No. 200021_138039. We thank Ulrich Krieger and Anand Kumar for fruitful discussion.

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

**Table 1.** Spearman's rank correlation coefficients for Hoggar Mountain dust samples (H1 – H12) and an ATD
10   sample (A5). Experiments with cooling rates 1 K/min and 10 K/min are flagged by "(1)" and "(10)", respectively.
The uncertainties represent a 68% confidence interval.

| Sample | Cooling rate (K/min) | Spearman Coefficient |
|---|---|---|
| H1 | 10 | -0.26 ± 0.06 |
| H2 | 10 | 0.33 ± 0.09 |
| H3 | 10 | -0.10 ± 0.07 |
| H4 | 1 | -0.06 ± 0.14 |
| H5 | 1 | 0.21 ± 0.09 |
| H6 | 1 | -0.48 ± 0.07 |
| H7 | 1 | -0.09 ± 0.09 |
| H8(1) | 1 | -0.40 ± 0.11 |
| H8(10) | 10 | 0.34 ± 0.09 |
| H9(1) | 1 | -0.30 ± 0.10 |
| H9(10) | 10 | 0.64 ± 0.09 |
| H10 | 1 | -0.81 ± 0.10 |
| H11 | 10 | -0.90 ± 0.07 |
| H12 | 10 | 0.39 ± 0.07 |
| A5 | 1 | 0.07 ± 0.16 |





**Table 2.** Spearman's rank correlation coefficient for birch pollen washing water samples P1 – P9. The uncertainties represent a 68% confidence interval.

| Sample | Cooling rate (K/min) | Spearman Coefficient |
|---|---|---|
| P1 | 1 | 0.18 ± 0.12 |
| P2 | 1 | 0.01 ± 0.09 |
| P3 | 1 | 0.07 ± 0.10 |
| P4 | 1 | 0.02 ± 0.08 |
| P5 | 1 | 0.15 ± 0.09 |
| P6(1) | 1 | 0.40 ± 0.13 |
| P6(10) | 10 | 0.02 ± 0.08 |
| P7(1) | 1 | -0.57 ± 0.09 |
| P7(10) | 10 | -0.06 ± 0.07 |
| P8 | 1 | 0.80 ± 0.12 |
| P9 | 1 | -0.65 ± 0.10 |



**Table 3.** CNT-based fits for Hoggar Mountain dust samples H1 – H9 and ATD sample A5 using the parameterizations Ick15, P&K97, and Z07 for version V1: fitting only contact angle $\alpha$ (°) and setting the prefactor $\beta = 1$. $A_{\text{crit,het}}$ is the calculated critical active site area in $\text{nm}^2$. All uncertainties correspond to 68% confidence intervals.

| Sample | Ick15 | | P&K97 | | Z07 | |
|---|---|---|---|---|---|---|
| | $\alpha$ | $A_{\text{crit,het}}$ | $\alpha$ | $A_{\text{crit,het}}$ | $\alpha$ | $A_{\text{crit,het}}$ |
| H1 | $43.9 \pm 0.3$ | 29.6 | $47.4 \pm 0.3$ | 33.3 | $39.5 \pm 0.2$ | 24.9 |
| H2 | $44.2 \pm 0.5$ | 31.7 | $47.5 \pm 0.5$ | 35.4 | $39.8 \pm 0.4$ | 26.7 |
| H3 | $42.7 \pm 0.2$ | 32.0 | $46.0 \pm 0.2$ | 36.0 | $38.4 \pm 0.2$ | 26.9 |
| H4 | $35.6 \pm 0.2$ | 46.3 | $38.2 \pm 0.2$ | 52.1 | $32.2 \pm 0.2$ | 38.8 |
| H5 | $42.8 \pm 0.2$ | 32.4 | $46.2 \pm 0.2$ | 36.5 | $38.6 \pm 0.2$ | 27.3 |
| H6 | $36.4 \pm 0.1$ | 44.8 | $39.0 \pm 0.2$ | 50.5 | $32.9 \pm 0.1$ | 37.6 |
| H7 | $41.8 \pm 0.2$ | 34.0 | $45.1 \pm 0.2$ | 38.3 | $37.7 \pm 0.1$ | 28.6 |
| H8 | $45.7 \pm 0.2$ | 26.3 | $49.2 \pm 0.2$ | 29.3 | $41.1 \pm 0.1$ | 22.2 |
| H9 | $44.0 \pm 0.3$ | 33.4 | $47.5 \pm 0.3$ | 37.6 | $39.7 \pm 0.2$ | 28.3 |
| A5 | $28.0 \pm 0.1$ | 80.5 | $29.9 \pm 0.1$ | 90.4 | $25.6 \pm 0.1$ | 67.8 |





**Table 4.** CNT-based fits for Hoggar Mountain dust samples H1 – H9 and ATD sample A5 using the parameterizations Ick15, P&K97, and Z07 for version V2: simultaneous fit of contact angle $\alpha$ (°) and prefactor $\beta$ (dimensionless). $A_{\mathrm{crit,het}}$ is the calculated critical active site area in nm$^2$.

| Sample | Ick15 | | | P&K97 | | | Z07 | | |
|---|---|---|---|---|---|---|---|---|---|
| | $\alpha$ | $10^5 \beta$ | $A_{\mathrm{crit,het}}$ | $\alpha$ | $10^5 \beta$ | $A_{\mathrm{crit,het}}$ | $\alpha$ | $10^5 \beta$ | $A_{\mathrm{crit,het}}$ |
| H1 | 36.5±0.7 | 0.38–1.196 | 21.7 | 40. 5±0.7 | 3.48–10.17 | 25.9 | 32.9±0.6 | 0.40–1.23 | 18.2 |
| H2 | 33.2±0.4 | 0.014–0.024 | 19.6 | 37.4±0.4 | 0.186–0.312 | 24.1 | 30.0±0.4 | 0.015–0.026 | 16.3 |
| H3 | 38.3±1.4 | 10.5–91.9 | 26.7 | 42.1±1.4 | 59–500 | 31.3 | 34.6±1.2 | 11.7–104.7 | 22.4 |
| H4 | 29.8±1.8 | 0.049–0.792 | 33.7 | 32.2±1.9 | 0.115–1.79 | 38.7 | 27.2±1.7 | 0.076–1.235 | 28.5 |
| H5 | 36.0±1.7 | 0.136–1.342 | 24.2 | 39.5±1.8 | 0.74–7.04 | 28.4 | 32.6±1.5 | 0.14–1.634 | 20.3 |
| H6 | 30.5±0.8 | 0.094–0.551 | 32.8 | 33.0±0.9 | 0.23–1.325 | 37.8 | 27.8±0.8 | 0.155–0.846 | 27.7 |
| H7 | 36.9±1.2 | 1.69–4.36 | 27.6 | 40.3±1.2 | 7–58 | 32.0 | 33.4±1.1 | 2.2–18.43 | 23.2 |
| H8 | 41.7±0.2 | 130–212 | 22.7 | 45.8±0.2 | 691–1102 | 26.3 | 37.6±0.1 | 142–220 | 19.0 |
| H9 | 36.5±0.9 | 0.209–0.901 | 24.5 | 40.5±0.9 | 2.01–8.09 | 29.2 | 32.9±0.8 | 0.231–0.987 | 20.5 |
| A5 | 20.8±0.4 | 2.92–8.22 | 45.7 | 22.2±0.4 | 4.93–11.79 | 51.8 | 19.1±0.3 | 4.95–12.03 | 39.0 |





**Table 5.** CNT-based fits for the birch pollen washing water samples P1 – P7 using the parameterizations Ick15, P&K97, and Z07 for version V1: fitting only contact angle $\alpha$ (°) and setting the prefactor $\beta = 1$. $A_{\mathrm{crit,het}}$ is the critical active site area in nm$^2$. For samples P6 and P7 refreeze runs carried out with 10 K/min (P6(10), P7(10) and 1 K/min (P6(1), P7(1)) are evaluated separately.

| Sample | Ick15 | | P&K97 | | Z07 | |
|---|---|---|---|---|---|---|
| | $A$ | $A_{\mathrm{crit,het}}$ | $\alpha$ | $A_{\mathrm{crit,het}}$ | $\alpha$ | $A_{\mathrm{crit,het}}$ |
| P1 | $46.31 \pm 0.08$ | 25.5 | $50.00 \pm 0.10$ | 28.6 | $41.63 \pm 0.07$ | 21.5 |
| P2 | $46.91 \pm 0.03$ | 27.5 | $50.68 \pm 0.04$ | 30.9 | $42.17 \pm 0.03$ | 23.3 |
| P3 | $42.62 \pm 0.02$ | 30.6 | $45.91 \pm 0.02$ | 34.4 | $38.36 \pm 0.02$ | 25.7 |
| P4 | $47.32 \pm 0.05$ | 26.1 | $51.11 \pm 0.06$ | 29.3 | $42.54 \pm 0.05$ | 22.1 |
| P5 | $47.40 \pm 0.03$ | 25.1 | $51.21 \pm 0.04$ | 28.1 | $42.61 \pm 0.02$ | 21.2 |
| P6(10) | $51.77 \pm 0.16$ | 18.2 | $55.74 \pm 0.22$ | 20.2 | $46.65 \pm 0.14$ | 15.6 |
| P6(1) | $50.93 \pm 0.17$ | 20.3 | $55.08 \pm 0.23$ | 22.7 | $45.82 \pm 0.15$ | 17.3 |
| P7(10) | $49.09 \pm 0.05$ | 22.4 | $53.04 \pm 0.08$ | 25.1 | $44.14 \pm 0.04$ | 19.0 |
| P7(1) | $46.88 \pm 0.10$ | 24.7 | $50.63 \pm 0.12$ | 27.7 | $42.14 \pm 0.08$ | 20.8 |
| P8 | $45.78 \pm 0.25$ | 26.4 | $49.42 \pm 0.31$ | 29.7 | $41.16 \pm 0.22$ | 22.3 |
| P9 | $47.19 \pm 0.12$ | 26.2 | $51.00 \pm 0.15$ | 29.4 | $42.42 \pm 0.10$ | 22.1 |





**Table 6.** CNT-based fits for birch pollen washing water samples P1 – P7 using the parameterizations Ick15, P&K97, and Z07 for version V2: simultaneous fit of contact angle $\alpha$ (°) and prefactor $\beta$ (dimensionless). $A_{\text{crit,het}}$ is the critical active site area in $nm^2$.

| Sample | Ick15 | | | P&K97 | | | Z07 | | |
|---|---|---|---|---|---|---|---|---|---|
| | $\alpha$ | $\beta$ | $A_{\text{crit,het}}$ | $\alpha$ | $\beta$ | $A_{\text{crit,het}}$ | $\alpha$ | $\beta$ | $A_{\text{crit,het}}$ |
| P1 | 49.6±3.3 | 920–67120 | 28.2 | 54.4±3.5 | 280–18610 | 32.2 | 44.4±2.9 | 53–4821 | 23.8 |
| P2 | 49.2±1.0 | 28–2 655 | 29.5 | 54.1±1.1 | 106–909 | 33.8 | 44.0±0.9 | 21.4–185.3 | 24.9 |
| P3 | 43.5±0.6 | 3.56–16.67 | 31.7 | 47.4±0.6 | 8.7–39.4 | 36.2 | 39.2±0.5 | 3.98–20.56 | 26.7 |
| P4 | 45.2±1.5 | 0.006–0.0855 | 24.3 | 49. ±1.6 | 0.05–0.543 | 28.3 | 40.5±1.3 | 0.0055–0.0665 | 20.4 |
| P5 | 48.4±1.0 | 2.33–20.53 | 25.9 | 53.4±1.1 | 10.9–90.2 | 29.8 | 43.4±0.9 | 1.62–13.94 | 21.8 |
| P6(10) | 64.5±3.5 | $(2.2–196)\times10^9$ | 24.1 | 72.3±4.0 | $(6.3–415)\times10^9$ | 26.8 | 56.82.9 | $(0.22–14.5)\times10^9$ | 20.6 |
| P6(1) | 68.2±3.5 | $(8.1–890)\times10^{15}$ | 29.1 | 76.2±4.1 | $(4.8–369)\times10^{15}$ | 31.8 | 59.9±2.9 | $(0.45–59)\times10^{15}$ | 25.3 |
| P7(10) | 51.9±1.2 | 47–518 | 24.3 | 57.5±1.3 | 287–2721 | 27.9 | 46.2±1.0 | 23.8–231.4 | 20.5 |
| P7(1) | 51.8±4.3 | 3400–326700 | 28.6 | 57.0±4.7 | 7300–769900 | 32.5 | 46.3±3.7 | 2000–197200 | 24.2 |
| P8 | 74.7±1.8 | $(6.2–513)\times10^{41}$ | 47.9 | 82.9±2.2 | $(2.8–249)\times10^{39}$ | 50.7 | 65.5±1.4 | $(1.3–111)\times10^{41}$ | 42.6 |
| P9 | 55.2±5.1 | $(4.5–838)\times10^6$ | 32.8 | 60.7±5.6 | $(6.6–1119)\times10^6$ | 37.0 | 49.2±4.3 | $(2.3–421)\times10^6$ | 27.9 |





**Table 7.** CNT-based fits for nonadecanol droplets N1 – N6 with radii $r = 31 - 1100$ µm measured by Zobrist et al. (2007) using the CNT-based parameterizations Ick15, P&K97, and Z07 for version V1: fitting only contact

5   angle $\alpha$ (°) and setting the prefactor $\beta = 1$. $A_{\text{crit,het}}$ is the calculated active site area in nm$^2$.

| Sample | Ick15 | | P&K97 | | Z07 | |
|---|---|---|---|---|---|---|
| | $\alpha$ | $A_{\text{crit,het}}$ | $\alpha$ | $A_{\text{crit,het}}$ | $\alpha$ | $A_{\text{crit,het}}$ |
| N1 ($r = 1100$ µm) | $40.3 \pm 1.0$ | 36.9 | $43.3 \pm 1.1$ | 41.6 | $36.3 \pm 0.8$ | 31.0 |
| N2 ($r = 1100$ µm) | $40.0 \pm 1.1$ | 44.3 | $43.0 \pm 1.2$ | 49.9 | $36.0 \pm 0.9$ | 37.2 |
| N3 ($r = 370$ µm) | $46.1 \pm 1.1$ | 28.2 | $49.8 \pm 1.2$ | 31.7 | $41.4 \pm 1.0$ | 23.8 |
| N4 ($r = 320$ µm) | $46.2 \pm 1.0$ | 32.0 | $49.9 \pm 1.1$ | 36.0 | $41.7 \pm 0.8$ | 27.2 |
| N5 ($r = 48$ µm) | $61.8 \pm 0.4$ | 13.9 | $66.1 \pm 0.5$ | 15.0 | $56.4 \pm 0.4$ | 12.4 |
| N6 ($r = 31$ µm) | $61.4 \pm 0.2$ | 13.1 | $66.0 \pm 0.2$ | 14.2 | $56.0 \pm 0.3$ | 11.7 |





**Table 8.** CNT fits for nonadecanol samples N1 – N6 with radii $r = 31 - 1100$ µm measured by Zobrist et al. (2007) using the parameterizations Ick15, P&K97, and Z07 for version V2: simultaneous fit of contact angle $\alpha$ (°) and prefactor $\beta$ (dimensionless). $A_{\text{crit,het}}$ is the calculated active site area in nm$^2$.

| Sample | Ick15 | | | P&K97 | | | Z07 | | |
|---|---|---|---|---|---|---|---|---|---|
| | $\alpha$ | $\beta$ | $A_{\text{crit,het}}$ | $\alpha$ | $\beta$ | $A_{\text{crit,het}}$ | $\alpha$ | $\beta$ | $A_{\text{crit,het}}$ |
| N1 | $27.9 \pm 0.5$ | $(1.2 - 2.0)\times10^{-8}$ | 19.3 | $31.1 \pm 0.7$ | $6.6$–$14.4\times10^{-8}$ | 23.6 | $25.3 \pm 0.5$ | $(1.2 - 2.5)\times10^{-8}$ | 16.1 |
| N2 | $27.1 \pm 0.9$ | $(0.6 - 1.5)\times10^{-8}$ | 22.3 | $30.2 \pm 0.9$ | $3.4 - 9.2\times10^{-8}$ | 27.2 | $24.6 \pm 0.8$ | $(0.7 - 1.9)\times10^{-8}$ | 18.7 |
| N3 | $32.7 \pm 0.4$ | $(3.2 - 5.0)\times10^{-8}$ | 15.9 | $37.4 \pm 0.7$ | $53$–$112)\times10^{-8}$ | 20.1 | $29.5 \pm 0.3$ | $(3.3 - 5.2)\times10^{-8}$ | 13.2 |
| N4 | $31.8 \pm 1.0$ | $(1.1 - 2.8)\times10^{-8}$ | 17.1 | $36.4 \pm 1.1$ | $18 - 52)\times10^{-8}$ | 21.6 | $28.7 \pm 0.8$ | $(1.1 - 3.0)\times10^{-8}$ | 14.2 |
| N5 | $61.0 \pm 4.8$ | $0.07$–$2.7$ | 13.7 | $71.5 \pm 5.2$ | $24$–$569$ | 16.1 | $53.3 \pm 4.0$ | $0.003$–$0.120$ | 11.5 |
| N6 | $58.8 \pm 2.0$ | $0.02$–$0.16$ | 12.5 | $69.2 \pm 2.3$ | $6$–$50$ | 14.9 | $51.5 \pm 1.6$ | $0.0013$–$0.0108$ | 10.4 |

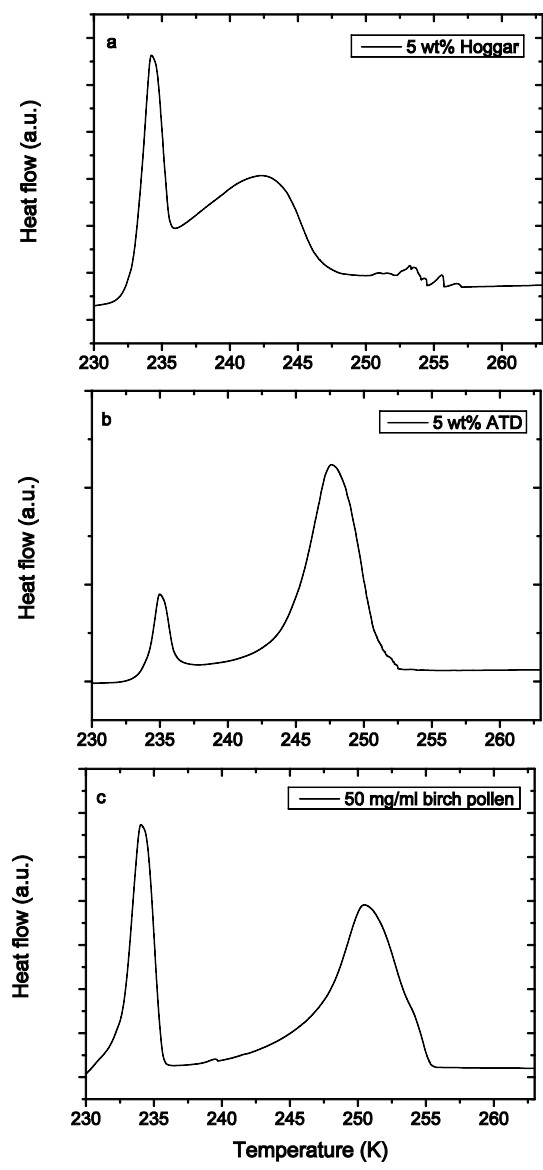

**Figure 1.** DSC thermograms of emulsion experiments for 1 K/min cooling rate. Panel a: Hoggar Mountain dust emulsion sample with homogeneous freezing peak at ~234 K and heterogeneous freezing peaking at ~244 K. The spikes around 253 K are due to latent heat release by a few very large water droplets containing Hoggar Mountain dust. Panel b: ATD emulsion sample. The broad peak at ~248 K stems from heterogeneous freezing induced by ATD particles present in many droplets, the narrow peak at ~235 K from homogeneous freezing of a few pure water droplets. Panel c: Birch pollen washing water emulsion sample with the heterogeneous freezing peak at ~250 K.





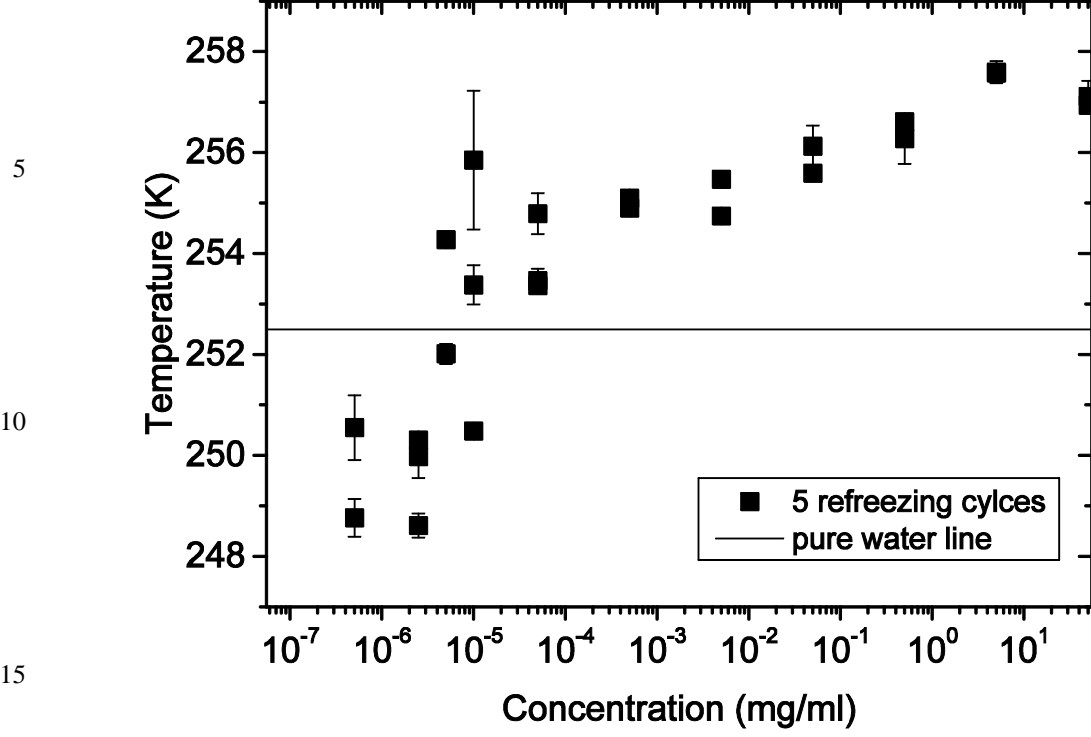

**Figure 2.** Dependence of freezing temperatures of birch pollen washing water on concentration. Symbols: mean freezing temperature of five freezing runs performed at a cooling rate of 10 K/min. Uncertainty ranges are given on the 68% confidence level. The concentration scale refers to the concentration of the birch pollen in suspension. Horizontal solid line: uppermost limit below which pure water samples may start to freeze.





**Figure 3**. Refreeze experiments with the Hoggar Mountain dust samples H1 – H12. Freezing onsets as a function of freezing run #. Squares: freezing runs with 10 K/min cooling rate. Circles: runs with 1 K/min. Filled symbols: 5 wt% samples. Open symbols: 0.5 wt% samples. Gray lines: bin intervals for runs with 1 K/min. Black lines: bin intervals for 10 K/min. Bin intervals are shown only for evaluated samples H1 – H9 (see text). Error bars given for the first data points are representative for all following data points acquired with the same cooling rate. They represent the instrumental temperature uncertainty as explained in Sect. 4.2. For 1 K/min runs the error bars are smaller than the symbol.





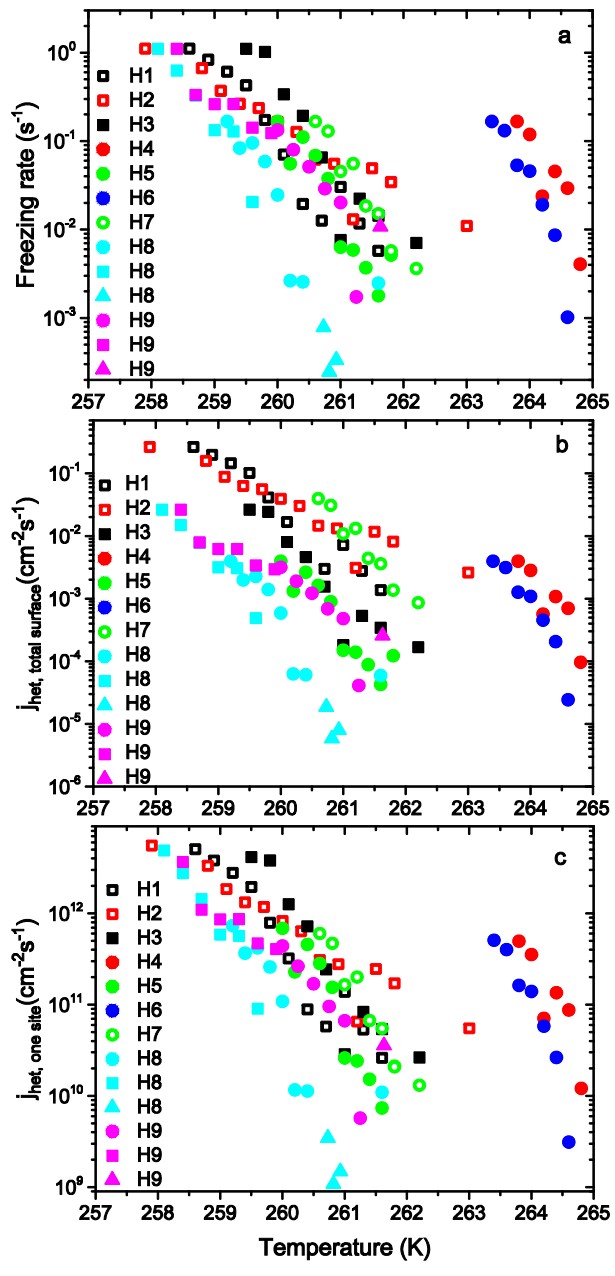

**Figure 4.** Hoggar Mountain dust samples prepared from 5 wt% (H3 – H6, H8, H9; filled symbols) and 0.5 wt% suspensions (H1, H2, H7, open symbols). Panel a: freezing rates. Panel b: Nucleation rate coefficients evaluated with respect to the total dust surface present in a sample as determined by BET measurements. Panel c: Nuclea-

30 tion rate coefficients with respect to the surface of one single site. Squares: 10 K/min cooling rate. Circles: 1 K/min cooling rate. Triangles: constant temperatures.





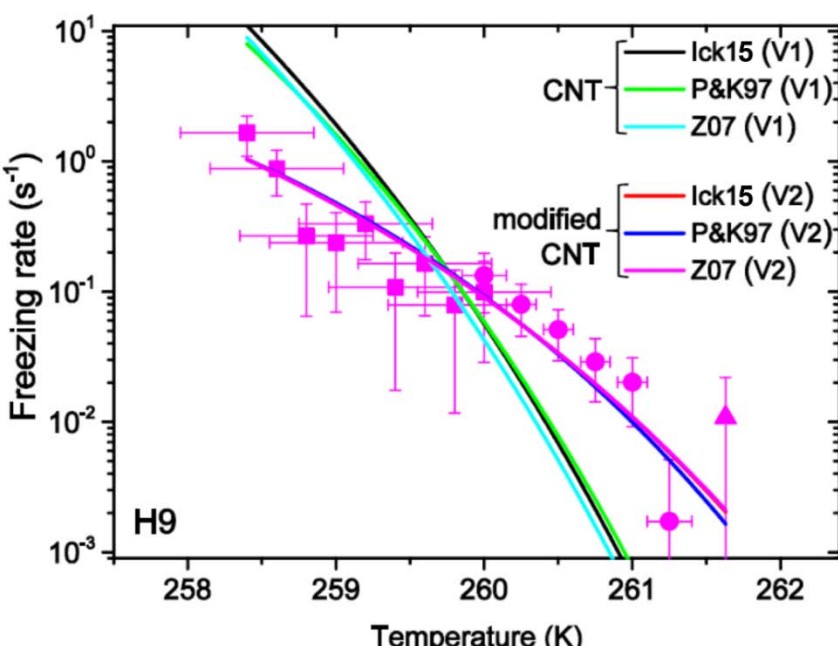

**Figure 5**. CNT-based fits of freezing rates for the Hoggar Mountain dust sample H9 with the parameterizations
5   by Ick15, P&K97 and Z07. Version V1: CNT fits performed with contact angle $\alpha$ as the only fit parameter. Version V2: modified CNT fits performed with $\alpha$ and $\beta$ as fit parameters. Note that Z07 (V2) mostly overlaps Ick15 (V2).



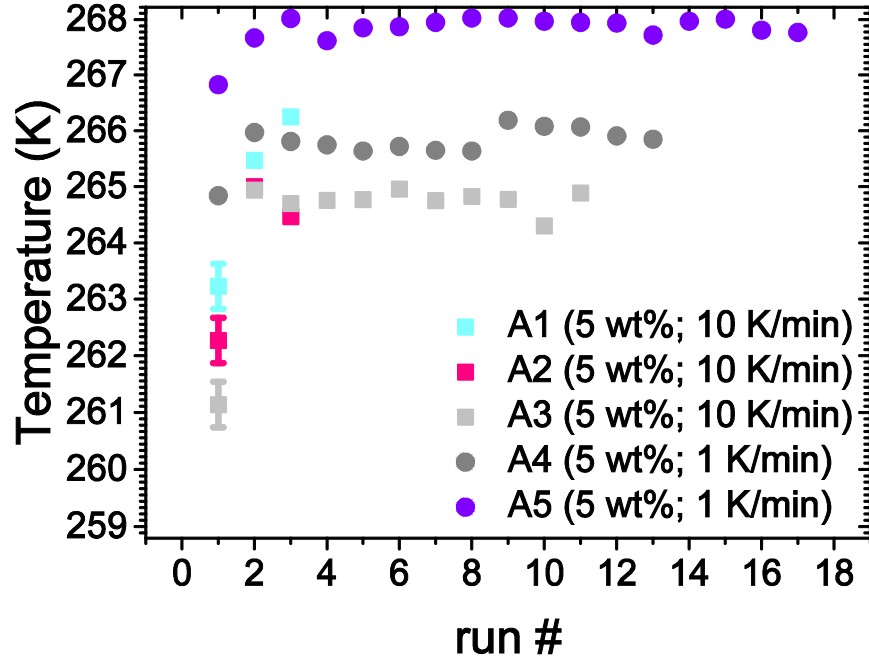

**Figure 6.** Refreeze experiments with the Arizona test dust (ATD) samples A1 – A5. Freezing onsets as a function of freezing run #. Squares: freezing runs with 10 K/min cooling rate. Circles: with 1 K/min cooling rate. All samples had a suspension concentration of 5 wt% and show a memory effect. Error bars given for the first data points are representative for all following data points acquired with the same cooling rate. They represent the instrumental temperature uncertainty as explained in Sect. 4.2. For 1 K/min runs the error bars are smaller than the symbols.





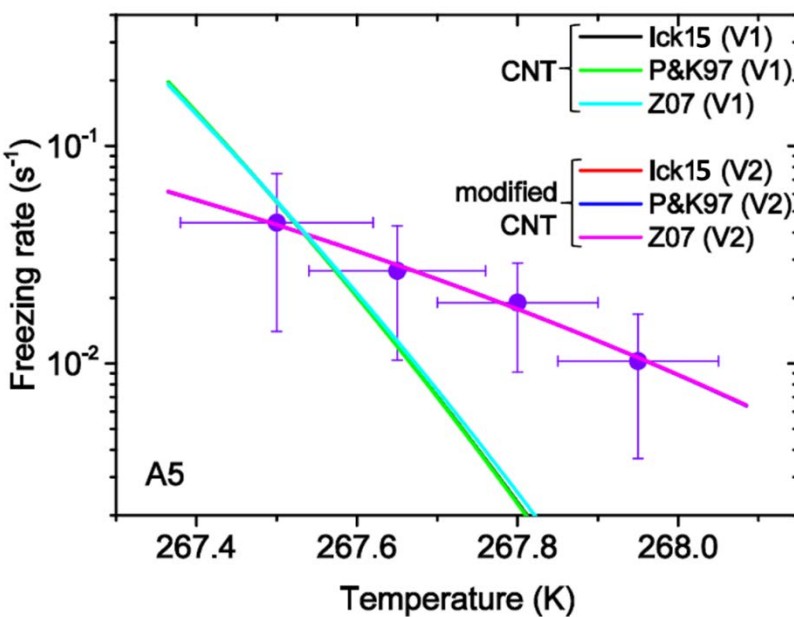

**Figure 7.** ATD sample A5 (suspension with 5 wt% dust). CNT-based fits of freezing rates for the parameterizations Ick15, P&K97, and Z07. V1: fits with contact angle $\alpha$ as the only fit parameter. V2: fits with $\alpha$ and $\beta$ as fit parameters. Fitting curves belonging to the same version are partly overlapping.



**Figure 8.** Refreeeze experiments with birch pollen washing water samples P1 – P9. Freezing onsets as a function of freezing run #. Filled symbols: 50 mg/ml samples. Half-filled symbols: 0.1 mg/ml samples. Open symbols: 0.001 mg/ml samples. Squares: freezing runs with 10 K/min cooling rate. Circles: runs with 1 K/min. Thin gray lines: bin intervals for runs with 1 K/min. Thick black lines: bin intervals for 10 K/min. Error bars given for the first data points of 10 K/min runs are representative for all following data points acquired with the same cooling rate. They represent the instrumental temperature uncertainty as explained in Sect. 4.2. For 1 K/min runs the error bars are smaller than the symbols.







**Figure 9.** Birch pollen washing water samples P1 – P9. Freezing rate (panel a) and freezing rate coefficients (panels b - d). Panel b: Lower limit of nucleation rate coefficients $j_{het}$ considering the whole surface of all macromolecules as ice nucleation active. Panel c: Nucleation rate coefficient $j_{het}$ considering a fraction of $10^{-7}$ of the macromolecules to be ice nucleation active. Panel d: Upper limit of nucleation rate coefficients $j_{het}$ with respect to the surface of one single active site. Filled symbols: 50 mg/ml samples. Half-filled symbols: 0.1 mg/ml samples. Open symbols: 0.001 mg/ml samples. Squares: 10 K/min cooling rate. Circles: 1 K/min cooling rate.



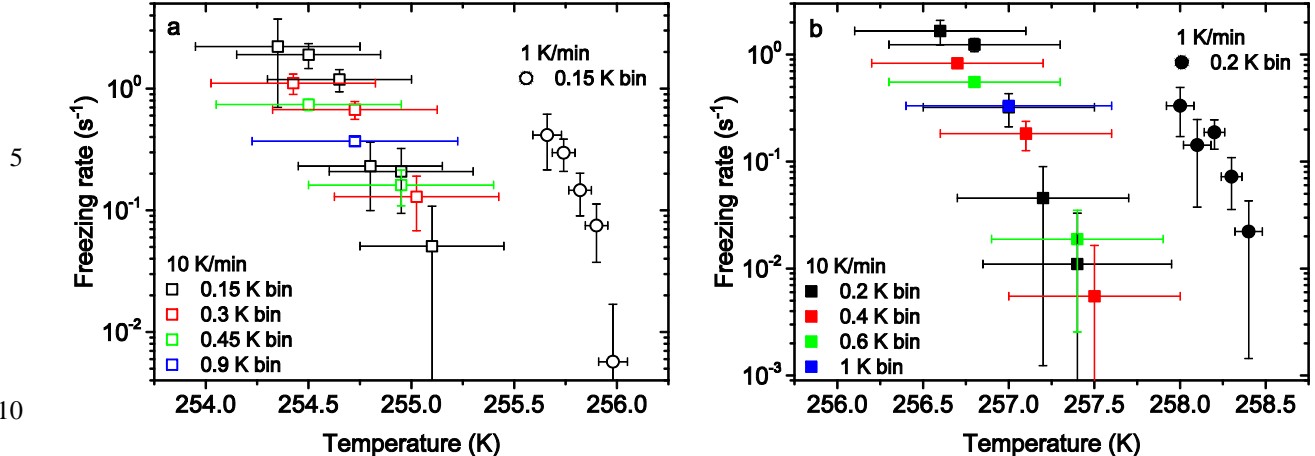

**Figure 10.** Freezing rates evaluated for Birch pollen washing water samples P6 (panel a) and P7 (panel b). Dependence of freezing rates on the choice of bin size for samples exposed to 10 K/min cooling rate (squares) and 1 K/min cooling rate (circles). Bin widths were varied between 0.15 K and 1 K (color coded). Horizontal error bars: temperature uncertainty within the droplet due to the precision of DSC temperature measurement. Vertical error bars: uncertainty due to the Poisson distribution.





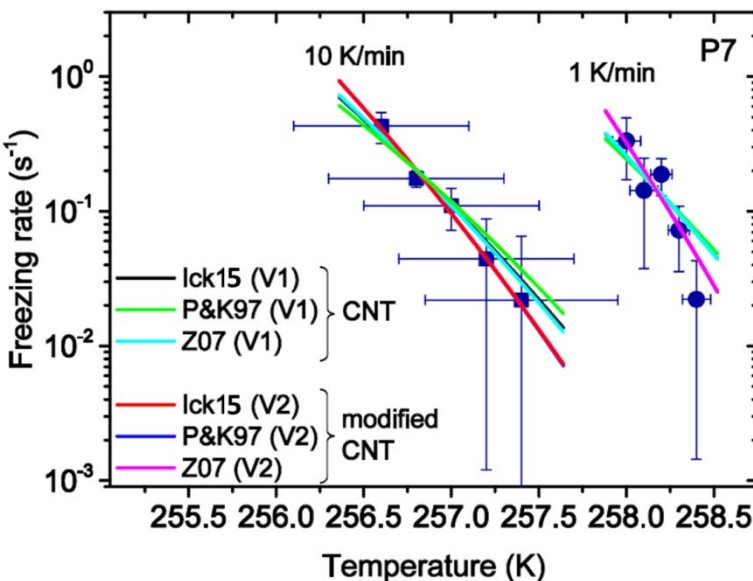

**Figure 11.** Birch pollen washing water sample P7 (50 mg/ml). CNT-based fits of freezing rates with the parameterizations Ick15, P&K97, and Z07. Left: Fits for a cooling rate of 10 K/min. Right: Fits for a cooling rate of 1 K/min. Version V1: fits performed with the contact angle $\alpha$ as the only fit parameter. Version V2: fits performed with $\alpha$ and $\beta$ as fit parameters. Fitting curves belonging to the same version are partly overlapping. Values of fit parameters are given in Tables 5 and 6.




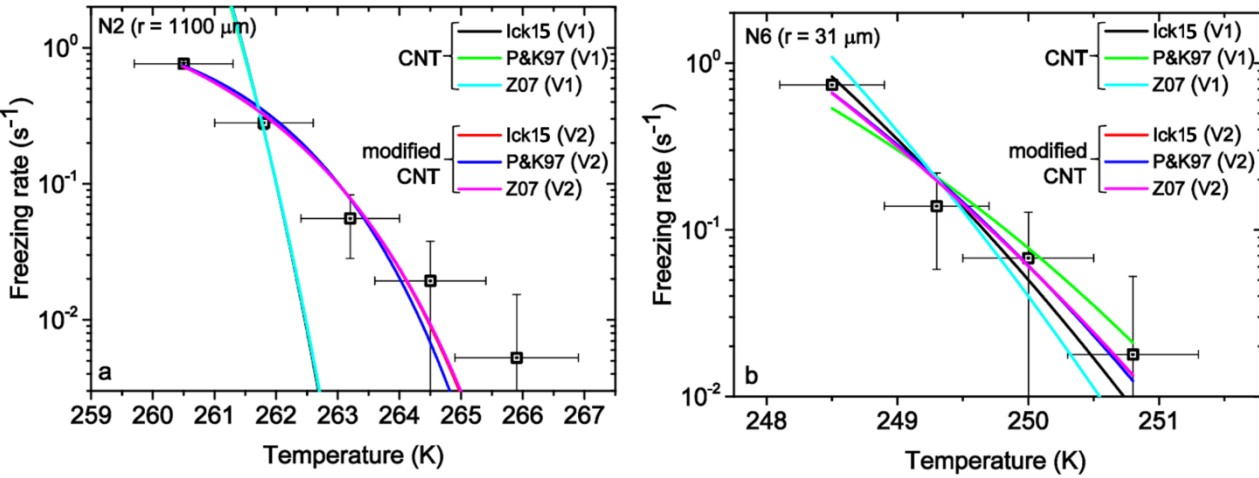

**Figure 12.** Nonadecanol samples N2 (large droplet, panel a) and N6 (small droplet, panel b). CNT-based fits of freezing rates measured by Zobrist et al. (2007) with the parameterizations Ick15, P&K97, and Z07. V1: fits with contact angle $\alpha$ as the only fit parameter. V2: fits with $\alpha$ and $\beta$ as fit parameters. V1 parameterizations are completely overlapping, V2 parameterizations overlap partly.





**Figure A1.** CNT-based fits of freezing rates for the Hoggar Mountain dust H1 – H8 samples with the parameter-izations Ick15, P&K97, and Z07. See Fig. 5 for H9. V1: fits with the contact angle $\alpha$ as the only fit parameter. V2: fits with $\alpha$ and $\beta$ as fit parameters. Values of fit parameters are given in Tables 3 and 4.





5  **Figure A2.** Birch pollen washing water samples P1 – P9. CNT-based fits of freezing rates with the parameterizations Ick15, P&K97, and Z07. V1: fits with the contact angle $\alpha$ as the only fit parameter. V2: fits with $\alpha$ and $\beta$ as fit parameters. Values of fit parameters are given in Tables 5 and 6.



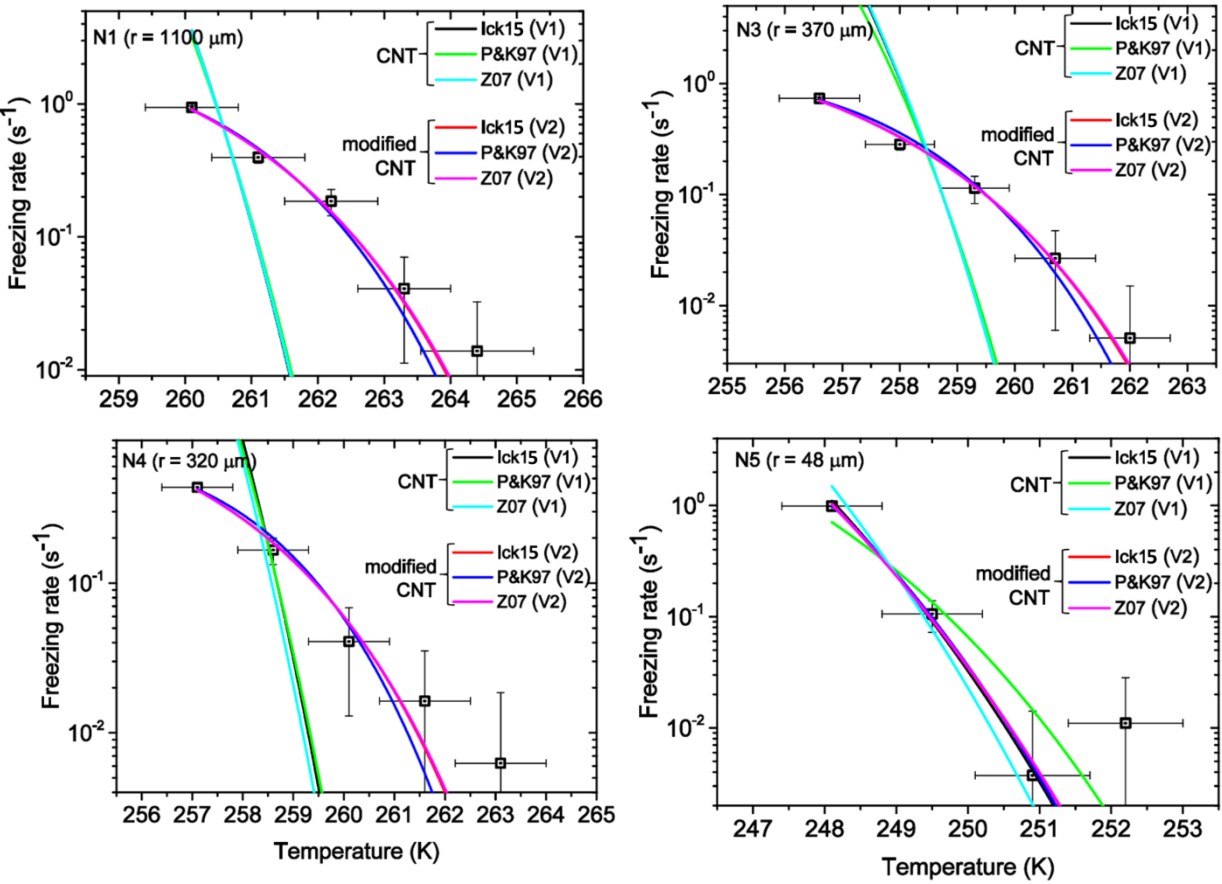

**Figure A3.** Nonadecanol samples N1, N3, N4, and N5 measured by Zobrist et al. (2007). CNT-based fits of freezing rates with the parameterizations Ick15, P&K97, and Z07. V1: fits with the contact angle $\alpha$ as the only fit parameter. V2: fits with $\alpha$ and $\beta$ as fit parameters. Values of fit parameters are given in Tables 7 and 8.

