# Peer review of "Refreeze experiments of water droplets containing different types of ice nuclei interpreted by classical nucleation theory"

_Atmospheric Chemistry and Physics, 2016_

## Referee Comment (RC1) · Anonymous Referee #1 · 3 Jan 2017

The paper by Kaufmann et al. deals with heterogeneous ice nucleation in the immersion freezing mode, which is an important parameter for climate issues. A systematic investigation has been presented focusing on the question whether an ice nucleus triggers a stochastic or a deterministic nucleation process. This question is not new but can be considered as still unresolved. The authors have chosen four well-known ice nuclei, i.e. two mineral dust samples (from Hoggar Mountains and Arizona Test Dust), a plant extract (birch pollen washing water) and a long-chain alcohol (nonadecanol). They use two fitting parameters to receive a parametrization, the contact angle ïĄą, which is the interfacial energy and the dimensionless pre-factor ïĄć, which is the deviation of the nucleation rate. Furthermore, they calculate the sizes of the nucleation

sites and the Spearman's rank correlation coefficient. The latter is determined from repeated freezing cycles and is used to discriminate between temperature and time dependence of heterogeneous ice nucleation, where temperature dependence counts for deterministic and time dependence for stochastic nucleation.

The paper is well-written and meets all criteria for a publication in ACP. However, there are a couple of revisions which should be carried out before publication:

Major Comments:

The concept of ice nucleation relies on the formation of stable hydrogen bonds and the formation of ice like structures which finally grow into a stable particle. In homogeneous ice nucleation only fluctuations and random structures impact this process and make it volume dependent. In heterogeneous ice nucleation the situation is more complicated since a substrate is catalyzing the phase transition. This substrate can be a macromolecule as well as a solid interface. The crucial point is that the rapid fluctuation of hydrogen bonds (life time 10ps) has to be stopped by interacting with the substrate. Up to now, nobody has ever observed morphology, structure and chemistry of an active nucleation site. All proof is of indirect nature. The authors should more carefully underline this fact at the beginning of the paper.

In general, I think that the samples are well-chosen, since they represent typical aerosol characteristics. However, the authors should compare only both dusts and both organic samples with each other – this demands a reorganization of some parts of the paper. From my point of view, it is highly questionable if a solid mineral surface and the surface of an organic and in water suspendable substance can exhibit comparable properties. The reason is that minerals offer steps, cracks and cavities at their surfaces which are active nucleation sites (see Kiselev A. et al (2016)), while the ice nucleation activity of organic macromolecules depends very much on their secondary, tertiary and/or quaternary structure. Folding and/or agglomeration of these rather large macromolecules can be very decisive and can change during repeated freezing cycles.

One should expect that most nuclei (minerals and organics) exhibit several different nucleation sites. In order to make these sites of different activity visible, one could measure highly diluted samples, which typically show a flattening of the freezing curve with a steplike structure. Each step counts for one kind of nucleation site. Evaluating the median freezing temperature of each step would give an idea of the number and activity of these sites and one could test for changes of these parameters in repeated freezing cycles. This might even be more informative than the overall parameters ïĄą and ïĄć. To make it clearer, in how far can the authors assure that assigned singular or stochastic properties are not the product of changes of the particle or the nucleation site itself? This would not only change typical singular freezing characteristics but also the stochastic freezing, which would interfere the whole concept of the here presented evaluation.

Specific Comments:

Introduction

When introducing the concepts of stochastic and deterministic nucleation, these papers should be quoted: Vali G. & Stansbury E.J. (1966), Bigg E.K. (1953), Vali G. (1971).

Evaluation

When introducing the dust samples it should come clearer what the differences between both mineral compositions are. A table would be very helpful listing the different crystalline phases and impurities and giving the respective compositions in percentage. Another important parameter is particle sizes and alterations of these particles. Zolles et al. (2015) have applied milling in order to generate new and fresh nucleation sites. They have also considered aging and blocking of these sites. It would be interesting to know in how far this also applies for the here used samples in repeated freezing cycles.

When introducing birch pollen washing water it should be stated clearly that the molecular identity of the macromolecules is still unknown. Pummer et al. (2012) have assumed polysaccharides or glycoproteins. Augustin et al. (2013) have already pointed out that birch pollen washing water has at least two distinctive ice nuclei within their broad organic mixture. The group of Koop (Dreischmeier et al. (2014)) has shown in their cryo-DSC studies that there really are two distinctive nuclei and that the ratio between both is changing with concentration. It would be interesting to know if the ratio between both these sites is changing in repeated freezing cycles and if both sites show the same character. When evaluating these results, the authors have worked with the assumption that the washing water consists solely of 300 kDa macromolecules. This has been assumed for modelling reasons only. However, 300 kDa is only the upper limit according to Pummer et al. and nothing is known about the real size distribution. It would be interesting to know how strong variations of the size would impact the results.

Minor comments

Often the x-axes of comparing figures are not scaled in the same way and make comparison difficult, e.g. figure 10 and all figures of the Appendix A.

Page 18: Nucleation rates should always be given with the respective temperature in parenthesis.

Page 22: The give number of active molecules should be accompanied by the respective sample volume (e.g. ml-1).

Figure 2: The volume of the sample should be given, since the pure water line is at rather high temperature, which is only understandable in the context of the respective sample volume.

When working with oil-water emulsions in repeating freezing cycles, one should carefully check for the stability of these emulsions. Further information can also be gathered from Hauptmann et al. (2016).

References

Bigg E.K. (1953) The supercooling of water. Proceedings of the Physical Society. Section B 9 66, 688-694.

Dreischmeier et al. (2014) Investigation of heterogeneous ice nucleation in pollen suspensions and washing water. Geophysical Research Abstracts, Vol. 16, EGU2014-8653, 2014

Hauptmann A. et al. (2016) Does the emulsification procedure influence freezing and thawing of aqueous droplets? Journal of Chemical Physics 145, 21; 211923-1 - 211923-14.

Kiselev A. et al (2016) Active sites in heterogeneous ice nucleation – the excample of K-rich feldspars. Science DOI: 10.1126/science.aai8034

Vali G. & Stansbury E.J. (1966) Time dependent characteristics of the heterogeneous nucleation of ice. Canadian Journal of Physics 44(3), 477-502.

Vali G. (1971) Quantitative evaluation of experimental results on heterogeneous freezing nucleation of supercooled liquids. Journal of the Atmospheric Sciences 28, 402-409.

Zolles T. et al (2015) Identification of ice nucleation active sites on feldspar dust particles, J. Phys. Chem. A, 119, 2692-2700, doi: 10.1021/jp509839x

---

## Referee Comment (RC2) · Anonymous Referee #2 · 5 Jan 2017

Kaufmann et al. present new heterogeneous ice nucleation data for water droplets that contain one of four substances: Hoggar Mountain dust, Arizona Test Dust, non-adeconol, and birch pollen washing water. Experiments are performed using differential scanning calorimetry. The authors use repeated freezing and thawing cycles while tracking the change in the onset freezing temperature with run number. These refreeze experiments are interpreted through the prism of classical heterogeneous nucleation theory to infer if freezing occurs quasi-deterministic on special active sites or stochastic on truly random locations on the surface.

The experimental setup has been used previously by the same authors and the experimental protocols are sound. Overall the manuscript is well written. The analysis

is interesting, informative, and relevant to the ACP readers. At times the theoretical modeling seems to push against the limits on what can be in inferred in principle using the adopted methodology. If properly caveated, I recommend publication.

Comments

The concept of onset freezing isn't very clear. The manuscript states: "For the refreeze experiments, the onset of the freezing peak was evaluated. The evaluation was done using the implemented software "TA Universal Analysis" of the instrument." First, this part needs to be expanded to explain how the procedure works, what threshold is used to detect onset. Perhaps this could be illustrated with the help of Figure 1? For the Hoggar dust, the first freeze events were dominated by large droplets. I presume they are only sometimes present and the configuration may be more unstable? Do these runs coincide with the ones that are flagged by the rank correlation? Second, would it be possible to estimate how many drops or how much water volume needs to freeze to detect onset? Since this is used to repeatably probe the same active site, i.e. the most efficient nuclei in the sample, it would be important to estimate many different sites are able to compete for the "onset" detection. If it is not possible to estimate this, the authors should comment how competition from different active sites would affect the run cycle statistics and derived nucleation rates.

Introducing the beta factor to improve fits of jhet seemed like a promising approach to gain additional insight into the mechanism of freezing. However, the results show variation of beta from 10ˆ-9 to 10ˆ43, or 52 orders of magnitude. The observable range of the universe is 10ˆ-15 to 10ˆ26 m, only 41 orders of magnitude. A physical interpretation of beta seems to stretch credulity.

As an optional suggestion. The nucleation rates for the different samples, e.g. Figure 4, could be normalized by 'active site' strength, assuming ones buys into that concept. One way to normalize the rates is to anchor rates at T0, where T0 corresponds to a nucleation rate of 10ˆ-3 cm-2 s-1 (middle panel of Figure 4). All other temperatures

would be relative to T0. The actual freezing temperature could be color coded onto the symbels. Such a plot might help to highlight similarities and differences in for the different active sites probed. If the graphs collapse, then a simple parameterization for a population of particles could be reported.

———————————————————

---

## Author Comment (AC1) · 14 Feb 2017

Responses to reviewer #1:

*We thank the reviewer for his/her thoughtful comments, which we address below in italic.*

The paper by Kaufmann et al. deals with heterogeneous ice nucleation in the immersion freezing mode, which is an important parameter for climate issues. A systematic investigation has been presented focusing on the question whether an ice nucleus triggers a stochastic or a deterministic nucleation process. This question is not new but can be considered as still unresolved. The authors have chosen four well-known ice nuclei, i.e. two mineral dust samples (from Hoggar Mountains and Arizona Test Dust), a plant extract (birch pollen washing water) and a long-chain alcohol (nonadecanol). They use two fitting parameters to receive a parametrization, the contact angle $\alpha$, which is the interfacial energy and the dimensionless pre-factor $c'$, which is the deviation of the nucleation rate. Furthermore, they calculate the sizes of the nucleation sites and the Spearman's rank correlation coefficient. The latter is determined from repeated freezing cycles and is used to discriminate between temperature and time dependence of heterogeneous ice nucleation, where temperature dependence counts for deterministic and time dependence for stochastic nucleation.

The paper is well-written and meets all criteria for a publication in ACP. However, there are a couple of revisions which should be carried out before publication:

Major Comments:

The concept of ice nucleation relies on the formation of stable hydrogen bonds and the formation of ice like structures which finally grow into a stable particle. In homogeneous ice nucleation only fluctuations and random structures impact this process and make it volume dependent. In heterogeneous ice nucleation the situation is more complicated since a substrate is catalyzing the phase transition. This substrate can be a macromolecule as well as a solid interface. The crucial point is that the rapid fluctuation of hydrogen bonds (life time 10ps) has to be stopped by interacting with the substrate. Up to now, nobody has ever observed morphology, structure and chemistry of an active nucleation site. All proof is of indirect nature. The authors should more carefully underline this fact at the beginning of the paper.

*We do this by adding the following sentence to the introduction:*

*", although direct evidence of the morphology, structure and chemistry of active nucleation sites is lacking up to now."*

In general, I think that the samples are well-chosen, since they represent typical aerosol characteristics. However, the authors should compare only both dusts and both organic samples with each other – this demands a reorganization of some parts of the paper. From my point of view, it is highly questionable if a solid mineral surface and the surface of an organic and in water suspendable substance can exhibit comparable properties. The reason is that minerals offer steps, cracks and cavities at their surfaces which are active nucleation sites (see Kiselev A. et al (2016)), while the ice nucleation activity of organic macromolecules depends very much on their secondary, tertiary and/or quaternary structure. Folding and/or agglomeration of these rather large macromolecules can be very decisive and can change during repeated freezing cycles.

*The manuscript in the present form treats the investigated ice nuclei each on its own in dedicated sections. Evidence for nucleation on active sites is discussed in Sect. 7.1.1 for Hoggar Mountain dust, in Sect. 7.1.2 for ATD, in Sect. 7.1.3 for birch pollen washing water and in Sect. 7.1.4 for nonadecanol coated droplets. The discussion of the prefactor $\beta$ is joined for the two dust samples in Sect. 7.3.1 because they are both mineral dusts and their $\beta$ -values are in the same range. Birch pollen washing water and nonadecanol coated droplets are treated in separate sections (7.3.2 and 7.3.3, respectively) because they represent different types of ice nuclei although they are both organic. The nonadecanol coating exhibits an extended surface and is in this respect comparable to the mineral dusts. However, the coating is supposed to be flexible and in this respect, it is comparable with the birch pollen macromolecules. The macromolecules are different from mineral dusts and the nonadecanol monolayer. We think that the manuscript organized like this is well suited to point out the nucleation characteristics of the different types of ice nuclei.*

One should expect that most nuclei (minerals and organics) exhibit several different nucleation sites. In order to make these sites of different activity visible, one could measure highly diluted samples, which typically show a flattening of the freezing curve with a steplike structure. Each step counts for one kind of nucleation site. Evaluating the median freezing temperature of each step would give an idea of the number and activity of these sites and one could test for changes of these parameters in repeated freezing cycles. This might even be more informative than the overall parameters $\alpha$ and $c'$. To make it clearer, in how far can the authors assure that assigned singular or stochastic properties are not the product of changes of the particle or the

nucleation site itself? This would not only change typical singular freezing characteristics but also the stochastic freezing, which would interfere the whole concept of the here presented evaluation.

*Although the samples that we used for the refreeze experiments contained high numbers of ice-nucleating sites, it is still possible to investigate single active sites when only one or a few sites are active at the highest temperature. In Sect. 7.1, we investigate the refreeze experiments for evidence against or in favor of ice nucleation on a single active site. We take sudden jumps of freezing temperature during refreeze experiments as "evidence that specific singular features in the samples are the nucleating entity, which might be fragile and can vanish or emerge during the course of a refreeze experiment" as stated in Sect. 7.1 . We show refreeze experiments with jumps and trends in Fig. 3 for Hoggar Mountain dust (H10 – H12) but do not evaluate them quantitatively to characterize active sites. We take significant differences in freezing temperatures between refreeze experiments as evidence for nucleation occurring on one or few active sites because such differences are very unlikely when a high number of nucleation sites are active at the same temperature. We state in Sects. 7.1.1 and 7.1.2 that in the case of Hoggar Mountain dust and ATD ice nucleation is likely to occur on the same best active site during the course of a refreeze experiment. For birch pollen washing water, we think that freezing occurs on the same best macromolecule for some samples. We state that for the nonadecanol coated samples no conclusion is possible based on the available experimental data.*

*We discuss in Sect. 7.3 the possibility that we misinterpret trends in the quality of a site as stochastic behavior. Under point (i) we state "If some sites were not constant in quality from one freezing cycle to the next, ice nucleation on such sites would not be fully stochastic. In this case, it would not be correct to describe the freezing temperature sequences with a constant contact angle $\alpha$. When variability of $\alpha$ is mistaken as random fluctuations of freezing temperatures, a low value of prefactor $\beta$ would be fitted. The presence of a monotonic trend or an improbable sequence of freezing temperatures are indications that nucleation indeed was not fully stochastic. However, there is no criterion available to discriminate stochastic variations of freezing temperature from variations due to variability of $\alpha$."*

Specific Comments:

Introduction

When introducing the concepts of stochastic and deterministic nucleation, these papers should be quoted: Vali G. & Stansbury E.J. (1966), Bigg E.K. (1953), Vali G. (1971).

*We added these references as suggested.*

Evaluation

When introducing the dust samples it should come clearer what the differences between both mineral compositions are. A table would be very helpful listing the different crystalline phases and impurities and giving the respective compositions in percentage. Another important parameter is particle sizes and alterations of these particles. Zolles et al. (2015) have applied milling in order to generate new and fresh nucleation sites.

*We discuss the composition of the dust samples later in the text (Sect. 7.1.1 for Hoggar Mountain dust and 7.1.2 for ATD) and refer to Kaufmann et al. (2016) and Atkinson et al. (2013) for the detailed compositions. Too much emphasis on the mineralogy could be misleading since the refreeze experiments probe the best sites present in the samples, while X-ray powder diffraction determines the average composition of the samples. We state in Sect. 7.1.2 that it is most probably the microcline component that nucleates ice in the refreeze experiments with ATD. The situation is less clear for Hoggar Mountain dust as discussed in Sect. 7.1.1.*

They have also considered aging and blocking of these sites. It would be interesting to know in how far this also applies for the here used samples in repeated freezing cycles.

*This is indeed an interesting question. We used the minerals without any preprocessing. A permanent blocking by impurities present in the samples from the beginning would remain undetected. A transient blocking could give rise to the jumps in freezing temperatures observed for some of the samples. Aging could give rise to the trends observed in some samples. We address these aspects more explicitly in Sect. 7.1.1 of the revised manuscript where we discuss the trends and jumps observed during the Hoggar Mountain dust refreeze experiments H10 – H12.*

When introducing birch pollen washing water it should be stated clearly that the molecular identity of the macromolecules is still unknown. Pummer et al. (2012) have assumed polysaccharides or glycoproteins.

Augustin et al. (2013) have already pointed out that birch pollen washing water has at least two distinctive ice nuclei within their broad organic mixture. The group of Koop (Dreischmeier et al. (2014)) has shown in their cryo-DSC studies that there really are two distinctive nuclei and that the ratio between both is changing with concentration. It would be interesting to know if the ratio between both these sites is changing in repeated freezing cycles and if both sites show the same character.

*We investigated the Czech birch pollen which is supposed to contain only one ice nucleation active macromolecule type. This precludes statements about the predominance of one or the other type. We investigated the macromolecules that are active at the highest temperatures, since only these come into action in bulk experiments.*

*We add a paragraph at the beginning of Sect. 7.1.3 to summarize what is known about the molecular identity of the macromolecules contained in birch pollen washing water:*

*"The molecular identity of the macromolecules in birch pollen washing water is still unknown. Pummer et al. (2012) suspected them to be polysaccharides or glycoproteins based on their resistivity against denaturation by 6 M guanidinium chloride and heating to 400 K. The ice nucleation activity differs slightly between birch pollen washing water from different geographical regions. Augustin et al. (2013) found that Swedish birch pollen washing water shows a second plateau in the temperature range between 249 and 256 K, which is absent in Czech birch pollen washing water. In the present study, we investigate Czech birch pollen washing water."*

When evaluating these results, the authors have worked with the assumption that the washing water consists solely of 300 kDa macromolecules. This has been assumed for modelling reasons only. However, 300 kDa is only the upper limit according to Pummer et al. and nothing is known about the real size distribution. It would be interesting to know how strong variations of the size would impact the results.

*The estimation of the ice-nucleation active fraction of macromolecules just gives the order of magnitude because of the rough assumptions it is based on (see Sect. 5.2.2). Assuming a molecular weight of 100 kDa instead of 300 kDa would just reduce the ice-nucleation active fraction by a factor of three. To make clear that $10^{-7}$ is only a rough estimate of the ice-nucleation active fraction of macromolecules, we write in the revised manuscript "in the order of $10^{-7}$" in Sect. 7.1.3.*

Minor comments

Often the x-axes of comparing figures are not scaled in the same way and make comparison difficult, e.g. figure 10 and all figures of the Appendix A.

*It is always difficult to choose the right scale for best comparisons between figures and to keep the features of interest within one figure visible. The comparison between panels (a) and (b) is not the focus of Fig. 10. The intension is to show that the bin width does not influence the freezing rates obtained for 10 K/min. This is why we chose different scales for the two panels. If we had used the same x-scale, one half of each panel would remain empty.*

*The focus of the figures in the Appendix A, is the quality of fit rather than the differences in the nucleation rate coefficients between the refreeze experiments. The comparisons of the refreeze experiments are given in Fig. 4 for Hoggar Mountain dust, in Fig. 6 for ATD and in Fig. 9 for birch pollen washing water. We would therefore prefer to keep the scales as they are.*

Page 18: Nucleation rates should always be given with the respective temperature in parenthesis.
*We add this information in the revised manuscript.*

Page 22: The give number of active molecules should be accompanied by the respective sample volume (e.g. ml-1).
*We add this information in the revised manuscript.*

Figure 2: The volume of the sample should be given, since the pure water line is at rather high temperature, which is only understandable in the context of the respective sample volume. When working with oil-water emulsions in repeating freezing cycles, one should carefully check for the stability of these emulsions. Further information can also be gathered from Hauptmann et al. (2016).

*We add this information to the figure caption of Fig. 2. The data in this figure is from bulk samples (large droplets) which we used for the refreeze experiments. The DSC thermograms in Fig. 1 are from emulsion experiments. We show them because they display the freezing due to average sites while bulk water freezing is due to nucleation on best sites. We do not show thermograms of the bulk experiments because the only information they provide is the onset temperature of the freezing peak, which is just one number.*

References

Bigg E.K. (1953) The supercooling of water. Proceedings of the Physical Society. Section B 9 66, 688-694.

Dreischmeier et al. (2014) Investigation of heterogeneous ice nucleation in pollen suspensions and washing water. Geophysical Research Abstracts, Vol. 16, EGU2014-8653, 2014

Hauptmann A. et al. (2016) Does the emulsification procedure influence freezing and thawing of aqueous droplets? Journal of Chemical Physics 145, 21; 211923-1- 211923-14.

Kiselev A. et al (2016) Active sites in heterogeneous ice nucleation – the example of K-rich feldspars. Science DOI: 10.1126/science.aai8034

Vali G. & Stansbury E.J. (1966) Time dependent characteristics of the heterogeneous nucleation of ice. Canadian Journal of Physics 44(3), 477-502.

Vali G. (1971) Quantitative evaluation of experimental results on heterogeneous freezing nucleation of supercooled liquids. Journal of the Atmospheric Sciences 28, 402-409.

Zolles T. et al (2015) Identification of ice nucleation active sites on feldspar dust particles,

J. Phys. Chem. A, 119, 2692-2700, doi: 10.1021/jp509839x

---

## Author Comment (AC2) · 14 Feb 2017

**Responses to reviewer #2:**

*We thank the reviewer for his/her suggestions, which we address below in italic.*

Kaufmann et al. present new heterogeneous ice nucleation data for water droplets that contain one of four substances: Hoggar Mountain dust, Arizona Test Dust, nonadeconol, and birch pollen washing water. Experiments are performed using differential scanning calorimetry. The authors use repeated freezing and thawing cycles while tracking the change in the onset freezing temperature with run number. These refreeze experiments are interpreted through the prism of classical heterogeneous nucleation theory to infer if freezing occurs quasi-deterministic on special active sites or stochastic on truly random locations on the surface.

The experimental setup has been used previously by the same authors and the experimental protocols are sound. Overall the manuscript is well written. The analysis is interesting, informative, and relevant to the ACP readers. At times the theoretical modeling seems to push against the limits on what can be in inferred in principle using the adopted methodology. If properly caveated, I recommend publication.

Comments

The concept of onset freezing isn't very clear. The manuscript states: "For the refreeze experiments, the onset of the freezing peak was evaluated. The evaluation was done using the implemented software "TA Universal Analysis" of the instrument." First, this part needs to be expanded to explain how the procedure works, what threshold is used to detect onset. Perhaps this could be illustrated with the help of Figure 1? For the Hoggar dust, the first freeze events were dominated by large droplets. I presume they are only sometimes present and the configuration may be more unstable? Do these runs coincide with the ones that are flagged by the rank correlation? Second, would it be possible to estimate how many drops or how much water volume needs to freeze to detect onset? Since this is used to repeatedly probe the same active site, i.e. the most efficient nuclei in the sample, it would be important to estimate many different sites are able to compete for the "onset" detection. If it is not possible to estimate this, the authors should comment how competition from different active sites would affect the run cycle statistics and derived nucleation rates.

*In Fig. 1, we show thermograms of emulsion experiments. However, we performed the refreeze experiments with single droplets of 1.8 – 2 mg weight (see Sect. 4.2). These droplets freeze immediately and completely due to the first nucleation event. The freezing is so abrupt that the sample cannot be cooled sufficiently to maintain the prescribed cooling rate and the temperature of the sample even increases instead of decreasing. The freezing onset of large droplets is therefore clearly given and taken as the nucleation temperature. To avoid confusion, we state in Sect. 4.2 of the revised manuscript: "Refreeze experiments were carried out with bulk samples which exhibit an abrupt heat release when they freeze leading in the DSC thermograms to a clear onset of the freezing peak, which was taken as the nucleation temperature."*

*Moreover, we improve Sect. 6.5, which presents the emulsion measurements. It now reads:*

*"In Fig. 1 typical thermograms of emulsion measurements with Hoggar Mountain dust (panel a), ATD (panel b) and birch pollen washing water (panel c) are shown. For ATD, Marcolli et al. (2007) showed that the observed range of heterogeneous freezing temperatures cannot be described by assuming the same contact angle for all ATD particles. Rather, the ice-nucleating sites of ATD particles are required to be of different qualities. Note, that the refreeze experiments were performed with single droplets weighing 1.8 – 2 mg which contain a high number of particles. The best nucleation sites probed in the refreeze experiments with bulk samples are active from 260 to 268 K, i.e. at distinctly higher temperatures than the average sites probed in the emulsion experiments which nucleate ice below 252 K. In contrast to the bulk measurements, no memory effect was observed for ATD emulsions. Hoggar Mountain dust is a mixture of various minerals which are nucleating ice at quite different temperatures (Pinti et al., 2012; Kaufmann et al., 2016) giving rise to the broad freezing signal starting below 257 K with the freezing of single large emulsion droplets as shown in panel (a). Again there is no overlap in freezing temperatures between the emulsion measurement and the refreeze experiments performed with large single droplets which froze from 258 to 265 K. With an onset of 255 K, the heterogeneous freezing peak of the emulsion made from the birch pollen washing water exhibits a clear overlap with the freezing temperatures observed for bulk measurements which indicates that the ice nucleation active macromolecules present in the birch pollen washing water contain quite uniform nucleation sites."*

Introducing the beta factor to improve fits of $j_{het}$ seemed like a promising approach to gain additional insight into the mechanism of freezing. However, the results show variation of beta from 10^-9 to 10^43, or 52 orders

of magnitude. The observable range of the universe is $10^{-15}$ to $10^{26}$ m, only 41 orders of magnitude. A physical interpretation of beta seems to stretch credulity.

As an optional suggestion. The nucleation rates for the different samples, e.g. Figure 4, could be normalized by 'active site' strength, assuming ones buys into that concept. One way to normalize the rates is to anchor rates at T0, where T0 corresponds to a nucleation rate of $10^{-3}$cm$^{-2}$s$^{-1}$ (middle panel of Figure 4). All other temperatures would be relative to T0. The actual freezing temperature could be color coded onto the symbols. Such a plot might help to highlight similarities and differences in for the different active sites probed. If the graphs collapse, then a simple parameterization for a population of particles could be reported.

*If we understand correctly, this evaluation would not be based on CNT but just rely on a normalization to a selected nucleation rate coefficient. Such a presentation would have the advantage to directly disclose the differences in slopes between the experiments. However, for this normalization, an arbitrary nucleation rate coefficient (e.g. $10^{-3}$ cm$^{-2}$s$^{-1}$ as suggested by the reviewer) has to be chosen and the nucleation temperature for this rate coefficient has to be determined for each refreeze experiment by interpolating or extrapolating the nucleation rate coefficients as a function of freezing temperature. Since the increase of the nucleation rate coefficients with decreasing temperature is not linear, the choice of the fitting function would again be arbitrary. In our approach, the fitting curve has a theoretical basis because it is prescribed by CNT. Moreover, the measured increase of nucleation rate coefficients with decreasing temperature can be compared with the increase predicted by CNT. CNT predicts for the mineral dust samples a too steep and for the birch pollen washing water a too shallow increase, which is reflected by $\beta < 1$ determined for the mineral dusts and $\beta > 1$ for birch pollen washing water. We do not give too much weight to the actual value of the prefactor, rather, we use it as an indicator for a steep or a shallow increase of the nucleation rate coefficients with decreasing temperature. To make this clearer, we add the following sentence to Sect. 7.3 of the revised manuscript:*

*"Note that the values fitted for $\beta$ range from $10^{-9}$ to $10^{43}$ for all refreeze experiments and show uncertainties of a factor of hundred for individual fits to refreeze experiments. This shows that the exact value of $\beta$ is not well constrained. Nevertheless, the $\beta$-value can be used as an indicator for a steeper ($\beta > 1$) or a shallower ($\beta < 1$) increase of nucleation rate coefficients with decreasing temperature than predicted by CNT."*

---

## Author Response (AR2)

**Co-Editor Decision: Publish subject to technical corrections** (28 Feb 2017) by Dr. Daniel Knopf

Comments to the Author:

Dear Authors,

I am happy to accept your manuscript for publication in Atmospheric Chemistry and Physics. Since you are reporting freezing rates, I believe it would be fair to mention the caveats regarding uncertainty in particle surface areas as discussed in Hartmann et al. (2016) and Alpert and Knopf (2016). This is independent of applied data set. Despite same weight percent a few small particles will have a much larger surface area than a single larger one acting as INPs. This surface area will likely also vary from droplet to droplet (in large droplets or in emulsions experiments, etc.).

I just suggest to mention this in one sentence, maybe, in the conclusions so the reader is aware of this issue and that it is being discussed. I don't think any more changes on the manuscript are necessary.

With best regards,

Daniel Knopf

*Response:*

*We have added a sentence that mentions the influence of the uncertainty in particle surface areas to the introduction and cite the references indicated by the co-editor.*

[revised manuscript text omitted]